

# An assessment of the disequilibrium of Alaskan glaciers

Daniel R. Otto[1], Gerard H. Roe[1], John Erich Christian[2]

[1] Department of Earth and Space Sciences, University of Washington, Seattle, WA, US
[2] Department of Geography, University of Oregon, Eugene, OR, US

*Correspondence to*: Daniel R. Otto (drotto@uw.edu)

**Abstract.** The finite response time of alpine glaciers means that glaciers will be in a state of disequilibrium in the presence of a climate trend. Using a simple model of glacier dynamics, we use metrics of glacier geometry to evaluate the present-day disequilibrium for a population of 5600 alpine glaciers in Alaska. Our results indicate that glaciers throughout the region are in a severe state of disequilibrium. We estimate that the median glacier has only undergone 27% of the retreat necessary to

achieve equilibrium with the present-day climate. In general, glaciers with smaller areas have smaller response times, and so are closer to equilibrium than large glaciers. Because much of Alaska's glacier area is contained in a few large glaciers that are far from equilibrium, and because the rate of warming has increased in the last ~50 years, the median equilibration weighted by area is only 16%. Our estimates are sensitive to uncertainty in response time and to the shape of the warming trend. Uncertainty is greatest for intermediate glacier response times but is small for glaciers with the smallest and largest response

times. Finally, we demonstrate that accounting for the increased rate of warming in the late-20th century is important for estimating glacier disequilibrium, whereas the shape of the warming trend in the early-20th century is less relevant. Our results imply substantial future glacier retreat is already guaranteed regardless of the trajectory of future warming.

## 1 Introduction

A glacier can be thought of as a reservoir of ice supplied by an input flux of snow accumulation and depleted by an output flux

due to the ablation of ice (i.e., all processes that remove mass). Within the glacier, ice flows to redistribute mass from where it accumulates to where it is lost. As such, it takes time (typically decades to centuries) for a glacier's geometry to adjust to a change in climate. Therefore, in the face of a continuous, ongoing warming, glaciers are always in a state of disequilibrium – playing catch up to the evolving climate. Even if the climate were to stop changing today, there would be a period of continued glacier retreat reflecting the remaining adjustment towards a new equilibrium.

The lag of the change in glacier length behind a change in climate, was an early target for modern theoretical studies of glacier response (e.g., Nye, 1960; 1965), and also for early numerical modelling studies (e.g., Budd and Jensen, 1975; Oerlemans, 1986; Huybrechts, 1989). Jóhannesson et al. (1989) proposed a simple expression based on glacier geometry that yields response times of several decades for all but the largest alpine glaciers, and subsequent studies have supported this range. Such

assessments of glacier response time underly the "*very high confidence*" of the Intergovernmental Panel on Climate Change





(Fox-Kemper et al., 2021) that glacier retreat will continue in coming decades – glaciers are currently still responding to the warming that has already been observed.

The concept of "*committed warming*" was introduced into climate science as a metric of climate disequilibrium: how much future warming would there be if $CO_2$ levels were suddenly fixed at today's values (e.g., Hansen et al., 1985; Wetherald et al., 2001)? An alternative definition is the future warming from past human activity (e.g., Armour and Roe, 2011). The first definition is a more direct measure of the dynamical disequilibrium, and we use an equivalent version in this study. The concept of committed future change has been introduced to glaciology (e.g., Dyurgerov et al., 2009; Goldberg et al., 2015; Christian et al., 2018; Hartl et al., 2021): given that glaciers are out of equilibrium with the current climate, what is the "*committed retreat*" even if the climate were to stop changing today? An assessment of the current state of glacier disequilibrium is important for interpreting and attributing the cause of observed glacier retreat. It is also important because the current dynamical state of our glaciers are the initial conditions of our numerical models projecting future glacier change.

This study is an extension of Christian et al. (2018) and Christian et al. (2022a). Christian et al. (2018) used a hierarchy of numerical and analytical models to demonstrate that the key factors controlling committed retreat are the glacier response time, the strength of the climate trend, and the glacier geometry. Christian et al. (2022a) applied a simple analytical model to analyse disequilibrium for the glaciers of the Cascades Range in Washington State. Here we evaluate glacier disequilibrium for alpine glaciers in Alaska. We take advantage of new datasets and update our methods to better represent the much wider range of glacier geometries in Alaska. We also focus on how the shape of the anthropogenic warming trend impacts glacier disequilibrium. Given the long response time of some Alaskan glaciers, and given an acceleration of anthropogenic warming in recent decades, our method indicates a severe disequilibrium (and so a large committed retreat) for many Alaskan glaciers. For example, we estimate that the median observed retreat of the largest glaciers (>250 km$^2$) is only ~15% of the equilibrium retreat were climate to stop changing today. In Sect. 2 we provide a more detailed description of glacier disequilibrium and the metrics we use to characterize it. Our population of Alaskan glaciers is described in Sect. 3, and in Sect. 4 our updated methods are applied to estimate the distribution of response times for our glacier population. In Sect. 5 we provide population distributions of the range of disequilibrium that we estimate, and for three different assumed anthropogenic warming scenarios. We also analyse how errors in our estimated input parameters might propagate to our answers. We conclude with a summary and a discussion of how our estimates might be refined in future work.

## 2 Glacier disequilibrium

The concepts of glacier disequilibrium and committed retreat are illustrated in Fig. 1 for two otherwise identical glaciers with response times of $\tau = 25$ and 75 yr. A glacier's response time, $\tau$, describes the characteristic timescale over which a glacier gains or loses mass at its terminus in response to a change in climate. Consider the onset of a linear warming trend in 1880.



For the sake of simplicity, we assume a constant prior climate and omit interannual variability. Natural interannual climate variability means that a glacier's length at any instant will rarely be in equilibrium with the long-term climate average. However, in this study we focus solely on the disequilibrium associated with the anthropogenic warming trend. One reason for this is that Roe et al. (2021) considered a wide range of synthetic, modelled, and reconstructed climate scenarios that included natural variability, and showed that post-industrial glacier disequilibrium is overwhelmingly associated with the anthropogenic forcing. Secondly, glacier disequilibrium due to natural variability will recover on its own and fluctuate around zero. At the onset of the trend, we can define a temperature anomaly, $T'(t)$ (Fig. 1a). One can calculate the glacier-length anomaly, $L'_{eq}(t),$ that would be in equilibrium with $T'(t)$, which is shown by the dashed lines in Fig. 1b. The actual length anomaly, $L'(t)$, (solid lines in Fig. 1b) is always less than $L'_{eq}(t)$: it takes time for a glacier's geometry to adjust to an anomaly in mass balance, as it tends toward a new equilibrium state. Therefore, in the case of an ongoing warming trend, glacier length will always be in a state of disequilibrium with the climate; and the larger $\tau$ is, the greater will be the degree of disequilibrium. In our synthetic temperature scenario, we choose a year (2020, here) for the warming trend to cease. Disequilibrium will persist, albeit diminishing, as the glacier asymptotes to a new equilibrium state (Fig. 1b).

Disequilibrium can be characterized by the metric of *committed retreat,* $|L' - L'_{eq}|$, which is the additional retreat after the warming trend ceases. It is the future retreat that is already guaranteed to occur, even if there is no more warming (e.g., Dyurgerov et al., 2009; Goldberg et al., 2015; Christian et al., 2018). At the onset of warming, glaciers are initially slow to adjust, but increase their rate of change until it matches that of the warming trend (Fig. 1b), and so asymptote to a state of constant committed retreat (Fig. 1c). Over the 140 yr trend in our synthetic example, the smaller-$\tau$ glacier gets fully spun up to the applied trend; the larger-$\tau$ glacier is still in the early phase of adjustment and so experiences an accelerating retreat throughout the warming period.

Committed retreat is not the only metric of disequilibrium. Christian et al., (2018) defined the *fractional equilibration* as $f_{eq} = L'(t)/L'_{eq}(t)$. At any given moment in time, it is the ratio of the actual observed retreat to the glacier's eventual retreat if the climate were to stop warming immediately. It is related to the committed retreat, which is equal to $L'_{eq}(t)(1 - f_{eq})$. Fractional equilibration is a useful measure because it allows the committed retreat to be directly estimated from the observed retreat and, unlike committed retreat, it provides a direct comparison between glaciers with different $\tau$. Moreover, because $f_{eq}$ is a ratio, some constant parameters cancel, circumventing uncertainty in some inputs.

Roe and Baker (2014; RB14) developed a model for glacier fluctuations linearized about some prescribed mean-state geometry, and that accurately emulates models of flowline ice dynamics (see RB14, Christian et al., 2018). In response to mass-balance anomalies, $b'$, length fluctuations are governed by a linear, third-order differential equation:





$$\left(\frac{d}{dt} + \frac{1}{\epsilon\tau}\right)^3 L' = \frac{\beta}{\epsilon^3 \tau^2} b'(t) \quad , \qquad (1)$$

where $\tau$ is the glacier response time, $\beta$ is a dimensionless geometric constant that affects glacier sensitivity, and $\epsilon = 1/\sqrt{3}$. In turn, $\tau = -H/b_t$ (see Johannesson et al., 1989), where $H$ is a characteristic thickness and $b_t$ is the net (negative) mass balance at the terminus; and $\beta = A_{tot}/(wH)$, where $A_{tot}$ is glacier area and $w$ is the width of the terminus zone.

The equilibrium length response to a step-change in mass balance, $\Delta b$, is given by the solution to Eq. (1) when $dL'/dt = 0$:

$$L'_{eq} = \beta\tau\Delta b \quad . \qquad (2)$$

We note that a glacier's transient response to a step-change in climate is sigmoidal rather than exponential in shape (e.g., RB14). It is important not to treat $\tau$ as an e-folding timescale. Doing so approximately halves estimates of disequilibrium relative to the true disequilibrium (Christian et al., 2018).

For a linear climate trend starting at $t = 0$, $b'(t) = \dot{b}t$, where $\dot{b}$ is a constant, and the analytic solution to Eq. (1) is:

$$L'(t) = \left[1 - \frac{3\,\epsilon\tau}{t}\left(1 - e^{-\frac{t}{\epsilon\tau}}\right) + e^{-\frac{t}{\epsilon\tau}}\left(\frac{t}{2\,\epsilon\tau} + 2\right)\right]\beta\tau\dot{b}t \qquad (3)$$

(Roe et al., 2017). In the limit of $t \gg \tau$, glacier disequilibrium asymptotes to $|L' - L'_{eq}| = 3\epsilon\tau^2\beta\dot{b}$. From this formula we see that $|L' - L'_{eq}|$ is greatest for larger $\tau$, and for larger $\dot{b}$. For $t > 3\tau$, the disequilibrium is within 5% of the asymptotic limit (e.g., Fig. 1c, orange line). In Sect. 4.3, we estimate that the area-weighted median $\tau = 50$ yr for our population of Alaskan glaciers. The implication is that many of Alaska's larger glaciers are currently far from the asymptotic disequilibrium limit, and are thus still in their early, accelerating phase of adjustment. For a linear trend $L'_{eq}(t) = \beta\tau\dot{b}t$ which, with Eq. (3), gives:

$$f_{eq}(t) = 1 - \frac{3\,\epsilon\tau}{t}\left(1 - e^{-\frac{t}{\epsilon\tau}}\right) + e^{-\frac{t}{\epsilon\tau}}\left(\frac{t}{2\,\epsilon\tau} + 2\right). \qquad (4)$$

We note that $f_{eq}$ depends only on $\tau$ and the duration of the trend: because $f_{eq}$ is a ratio, both $\beta$ and $\dot{b}$ cancel. This cancelation occurs even when the trend is not constant, because of the linear nature of Eq. (1). We evaluate the impact of other trend shapes on $f_{eq}$ in Sect. 5.2. The dependence of $f_{eq}$ on only one parameter, $\tau$, makes it useful for assessing disequilibrium across a population of glaciers because it reduces the number of uncertain inputs.

Figure 1d shows $f_{eq}$ for our synthetic linear trend. It remains less than 0.1 for $t < \tau$, and increases most rapidly between $\tau$ and $3\tau$, as the glacier asymptotes to constant $|L' - L'_{eq}|$. Thereafter, $f_{eq}$ continues to increase, but rather more slowly, as the absolute disequilibrium becomes a progressively smaller fraction of the total retreat. Figure 1 illustrates the importance of $\tau$ in determining future glacier behavior. The absolute retreats of our two glaciers are quite similar in 2020, but their different response times ($\tau = 25, 75$ yr) means there are large differences in committed retreat (~3 km) and fractional equilibration (~40%): the glaciers are in different phases of their adjustment.

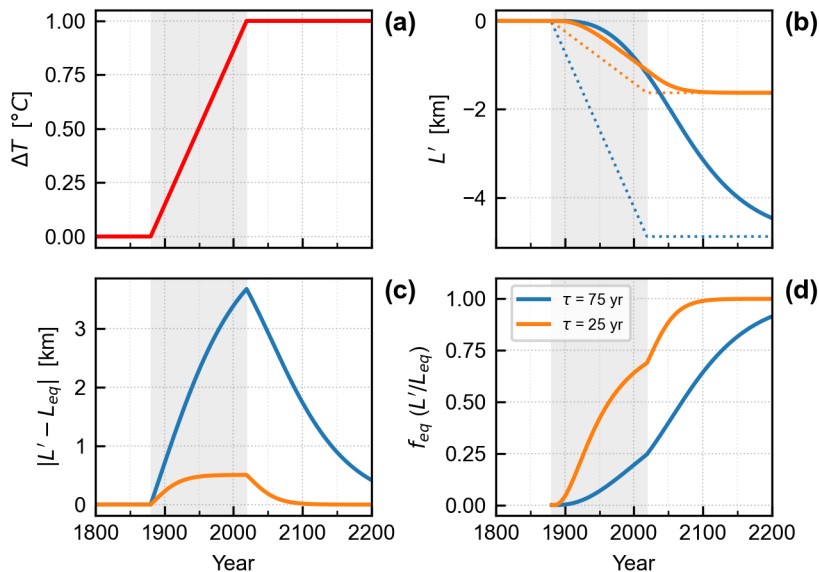

**Figure 1. (a) a constant warming trend initiated in 1880 that ceases in 2020 (b) change in glacier length, $L'$ (solid) and change in equilibrium length, $L_{eq}$ (dashed); (c) committed retreat; and (d) fractional equilibration, $f_{eq}$, for two otherwise-identical glaciers with response times of $\tau = 75$ yr (blue) and $\tau = 25$ yr (orange). The shaded area indicates the period of warming from 1880 to 2020. By 2020, the committed retreat is close to its asymptotic limit for the $\tau = 25$-yr glacier, while retreat is still accelerating for the $\tau = 75$-yr glacier. The curves were calculated using Eq. (1), with $\beta = 100$ (unitless) and $\Delta b = \mu\Delta T$, where the melt factor $\mu = 0.65$ m yr$^{-1}$ C$^{-1}$.**

## 3 Glacier inventory

We use the Randolph Glacier Inventory v6.0 (RGI, 2017), which provides metrics of glacier geometry, including length, maximum elevation, minimum elevation, area, and hypsometry. RGI calculates these statistics from glacier outlines, which reflect the glacier's state at a particular point in time. For RGI v6.0, outlines for Alaska were collected between 2004 and 2010. In this work we exclude: i) tidewater glaciers, ii) glaciers with reported areas less than 1 km², and iii) glaciers with less than 250 m of elevation change. Tidewater glaciers have complex terminus dynamics that can complicate their response to climate trends. Lastly, the ice dynamics of smaller, flatter glaciers are less likely to be well captured by our model of glacier response (e.g., Sanders et al., 2010), though their response to climate trends is certainly deserving of study. Of the 27,108 glaciers in Alaska delineated by RGI, 5,681 have areas greater than 1 km². Of the glacier area we exclude, 11,600 km² is from marine-terminating glaciers, 6400 km² is from glaciers with area less than 1 km², and 27 km² from the remaining glaciers that have





less than 250 m of elevation change. After applying our criteria, 5,205 glaciers remain for this analysis, which represents 77% of Alaska's glacier area. Hereafter, our analysis refers to this subset of glaciers.



**Figure 2. (a) Map of glacier area in Alaska, for glaciers analyzed in this study. The dot color shows the total glacierized area within each grid cell, as documented in the RGI v6 (RGI, 2017). Open circles indicate the location of long-term mass-balance records. The inset map shows the boundaries of secondary regions within Alaska as classified by the RGI: the Brooks Range (brown), Alaska Range (pink), Aleutian Range (red), Western Chugach Range (green), Wrangell and St. Elias Ranges (orange), and Coast Range (blue); (b) histogram of glacier numbers, by area; and (c) Distributions of total glacier area within each area bin, which indicates relative weights to each bin when results are area weighted.**

Figure 2a shows the geographic distribution of this subset, subdivided into the six glacierized sub-regions classified within the RGI. The histogram of glacier area shows that this population of glaciers is heavily skewed towards smaller glaciers (Fig. 2a).





A small number of large glaciers contribute much of the total area: half the total area comes from the largest 1.5% of the
glaciers. In our analysis below we choose to present both probability density functions (PDFs) and cumulative distribution
functions (CDFs). While the PDFs and the CDFs contain the same information, we find it useful to see both visual
representations. We also present these distributions both as simple glacier count and also area weighted. For the PDF of a
general glacier property, $x$, the area-weighted count, $n^w$, for the histogram bin with value $x_i$, is given by $n^w(x_i) =$
$(1/A_{tot}) \sum_j n_j(x_i) A_j$, where the index $j$ refers to all glaciers, with area $A_j$, that have value $x_i$, and $A_{tot}$ is the total glacier area.
The weights given to individual area bins can thus be seen in Fig. 2c, which shows how the total glacierized area is distributed
across the range of area bins. The larger glaciers contribute strongly to the area-weighted distributions, even though their
number is relatively few. In the appendix, we provide more details of how our analyses vary with glacier size by defining five
different area categories (Table A1).

## 4 Estimating glacier response time

### 4.1 Thickness, *H*

The ice thickness in the lower portion of the glacier is the most relevant for the dynamics governing advance and retreat (e.g.,
RB14). Christian et al. (2022a) estimates thickness for Cascadian glaciers using the method described in Haeberli and Hoelzle
(1995) that assumes a uniform slab geometry and a critical shear stress. Given Alaska's larger glaciers, we here choose a
method that can better represent thicknesses in the lower portion of the glacier.

We use the recently developed thickness dataset from Millan et al. (2022), which estimated thickness based on satellite
observations of ice velocity. We choose Millan et al. (2022) for our main analyses because they report that their method
exhibits less bias than the dataset of Farinotti et al. (2019) when compared to observed glacier thickness in Alaska. We report
the impact of using other thickness datasets on our analyses in Sect. 5.3. We take *H* to be the mean thickness along the central
flowline below the glacier's equilibrium-line altitude (ELA), with the flowline provided from the Open Global Glacier Model
(OGGM; Maussion et al., 2019). The glacier's ELA is estimated using RGI hypsometry and an assumption that the
accumulation-area ratio (AAR) is 0.6, which McGrath et al. (2017) found to be a plausible median for Alaskan glaciers.
Although the present-day AAR is estimated to be lower (e.g. Zeller et al., 2023), this is likely to be, at least in parts, a transient
consequence of the current disequilibrium. Because we aim to estimate $\tau$ prior to the onset of the warming trend (Fig. 1), 0.6
is a more appropriate value. We analyse the sensitivity of our results to these assumptions in Sect. 5.3.



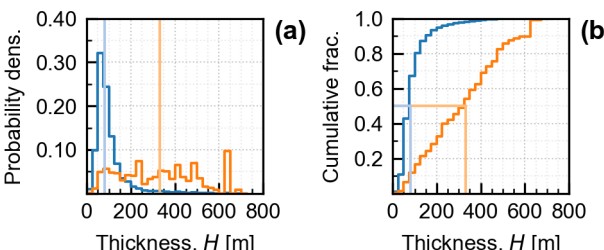

**Figure 3. (a) PDF and (b) CDF of estimated characteristic thickness ($H$) using the method outlined in the main text. Blue lines are number-weighted and orange lines are area-weighted. Lighter vertical lines indicate the medians of each distribution.**


The PDF and CDF of $H$ are presented in Fig. 3a and Fig. 3b, weighted by number (blue) and by area (orange). Weighted by number, the distribution of $H$ is characterized by a median (and 90% range) of 81 (42, 230) m. Weighted by area, the equivalent distribution has a median (and 90% range) of 331 (70, 644) m, which shows the substantial skew towards larger glaciers. Table

A1 shows a further breakdown of $H$ by area category. In general, the smallest values of $H$ come from small glaciers on steep slopes such as Williams Glacier ($H = 36$ m) and Byron Glacier ($H = 56$ m). The largest values of $H$ are associated with valley glaciers that have termini on a shallow slope. Examples include Tana Glacier ($H = 700$ m), Gilkey Glacier (473 m), and Kahiltna Glacier ($H = 448$ m).

Glacier thickness remains a major source of uncertainty in glaciology. Farinotti et al. (2017, 2019, 2021) report substantial variations among different estimation methods, with an approximate standard error of 50% when compared with observations (Farinotti et al., 2019). Nonetheless, for the purposes of evaluating a large network of glaciers, these geometry-based estimates capture the distribution of characteristic thicknesses well enough to be applied in our analysis.

**4.2 Terminus balance rate, $b_t$**

We calculate a glacier's terminus mass balance rate, $b_t$, by assuming a vertical mass-balance gradient, $db/dz$, that is constant over the ablation area delineated by its estimated ELA. We estimate a representative $db/dz$ for our population of glaciers from a compilation of point mass-balance observations of Alaskan glaciers. We include glaciers with observations spanning a sufficient elevation range to evaluate a balance profile, with particular emphasis on sufficient data in the ablation zone.


The compilation includes point mass-balance records from glaciers monitored by the USGS Benchmark Glacier Project, the Juneau Icefield Research Program, and other publications on selected well-observed glaciers (see Table B1). The locations of these glaciers are marked in Fig. 2a. While Alaska has relatively few mass-balance observations relative to its number of





glaciers, each of the state's glacierized regions are represented in our compilation. Figure 4 shows the vertical profile of mass

balance of each glacier averaged over the available record. Points denote the average mass balance and average elevation of

long-term measurement sites, and the associated solid lines are the least-squares fit of the evaluated data. Figure 4 shows our

assumption of a linear balance profile is a good approximation over the ablation zone. On the basis of this compilation, we

adopt a value for $db/dz$ of -6.5 m yr⁻¹ km⁻¹ (black dashed line in Fig. 4) as a reasonable central estimate for the region. The

calculated $db/dz$ values and details about the data evaluated for each glacier can be found in Table B2.

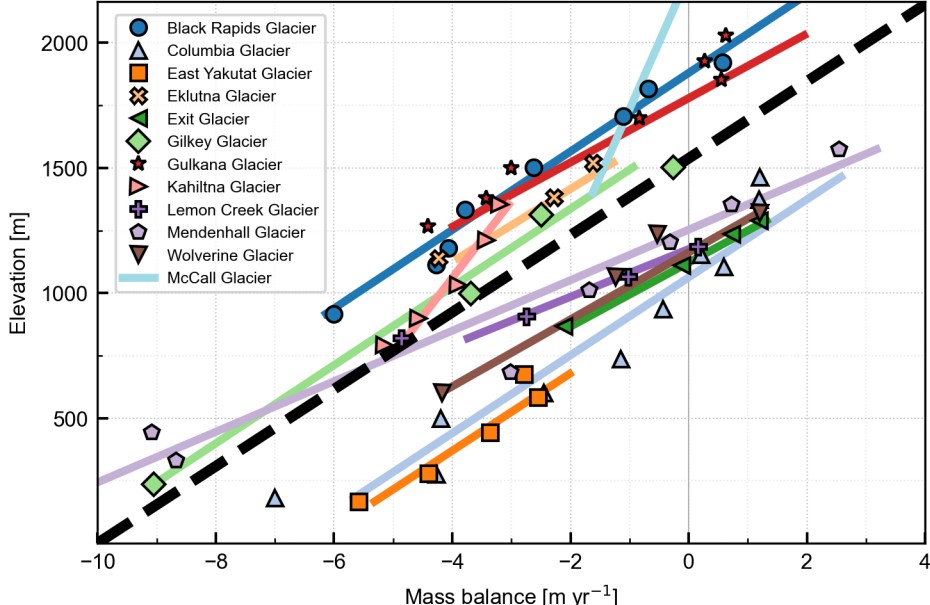


**Figure 4. Observations of the change in annual mass balance with elevation ($db/dz$). Points are the average values of long-term measurement sites. Colored lines are the least-squares fit. The value of $db/dz$ used for this analysis is shown by the dashed black line with a slope of 6.5 m yr⁻¹ km⁻¹. Data for McCall glacier is the mean profile combining intensive survey years 1969-1972 and 1993-1996 (Rabus and Echelmeyer, 1998). Marine-terminating glaciers are included here for context, though not in the rest of the**
**analysis. Data sources are listed in Table B1.**

Mass-balance profiles are known to differ by region due to regional differences in climatology, and among individual glaciers

due to local topography and geometry (Oerlemans and Hoogendoorn, 1989; Kaser, 2001; Benn and Lehmkuhl, 2000; Larsen

et al., 2015). However, we did not find statistically significant relationships between mass balance and continentality, slope,

aspect, or latitude; and so do not adjust $db/dz$ based on these factors. Although such relationships have been observed and

have a physical basis (Machguth et al., 2006; McGrath et al., 2015; McNeil et al., 2020; Olsen and Rupper, 2019; Florentine

et al., 2020), quantifying them with confidence for all of Alaska is precluded by the small number of glaciers with mass-

balance records. Further work on this topic is an important target of future research.



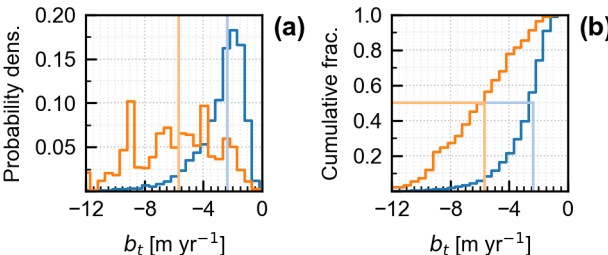

**Figure 5. (a) PDF and (b) CDF of estimated terminus balance rates ($b_t$). Blue lines are number-weighted and orange lines are area-weighted. Lighter vertical lines indicate the medians of each distribution. Like $H$, the area-weighted distribution is skewed towards larger (more negative) values.**

The PDF and CDF of calculated values for $b_t$ are presented in Fig. 5a and Fig. 5b, weighted by number (blue) and by area (orange). Weighted by number, the distribution of $b_t$ has a median (and 90% range) of -2.4 (-6.0, -1.0) m yr$^{-1}$. Weighted by

area, the equivalent distribution has a median (and 90% range) of -5.7 (-10.2, -1.9) m yr$^{-1}$. The difference in shape between the area-weighted and unweighted CDFs results from larger glaciers typically spanning a larger elevation range.

The most negative values of $b_t$ belong to glaciers with the largest elevation ranges, $dz$ (Table A1). For example, Logan Glacier ($b_t$ = -9.4 m yr$^{-1}$; $dz$ = 5145 m), Fairweather Glacier ($b_t$ = -8.8 m yr$^{-1}$; $dz$ = 4675 m), and Matanuska Glacier ($b_t$ = -8.7 m yr$^{-1}$;

$^{-1}$; $dz$ = 3408 m). Glaciers with elevation ranges greater than 2500 m can be found in the Wrangell-St. Elias Mountains, Alaska Range, and Chugach Mountains. Large maritime glaciers from the icefields of the Coast Range in Southeast Alaska span elevation ranges less than 2500 m but extend down to sea level and have similarly large magnitudes of $b_t$ (e.g., Meade Glacier: $b_t$ = -8.8 m yr$^{-1}$; $dz$ = 2193 m). Glaciers with the least negative values of $b_t$ tend to be found on the dry side of rain shadows (usually north-facing) and throughout the Brooks Range, where lower accumulation rates constrain a glacier's downslope

extent. Examples include the north-facing Chamberlin Glacier ($b_t$ = -2.5 m yr$^{-1}$) in the Brooks Range and Raven Glacier ($b_t$ = -2.8 m yr$^{-1}$) in the western foothills of the Chugach. Less negative values of $b_t$ also reflect glaciers which have already retreated upslope such as Ptarmigan Glacier ($b_t$ = -1.6 m yr$^{-1}$) and Flute Glacier ($b_t$ = -1.7 m yr$^{-1}$).

## 4.3 Glacier response time, $\tau$

We calculate $\tau$ for each glacier using our estimates of $H$ and $b_t$. Figure 6 shows the PDF and CDF of $\tau$. Weighted by number, $\tau$ has a median (and 90% range) of 35 (14, 96) yr. Weighted by area, the distribution has a median (and 90% range) of 52 (19, 167) yr. In general, larger area is associated with both larger $H$ and larger $b_t$ (Table A1). These have offsetting tendencies on $\tau$, so the range of estimated $\tau$ is narrower than might be expected given the order-of-magnitude variations in both $H$ and $b_t$. However, the fractional changes in $H$ are larger than the fractional changes in $b_t$, so the overall effect is a tendency for $\tau$ to

increase with glacier area (Table A1). Although the variance among individual glaciers is high, a linear regression of log(Area)



vs. log($\tau$) shows for every doubling of area, $\tau$ increases by a factor of 1.15. (Fig. C1; $R^2 = 0.09$) for glaciers with areas greater than 5 km$^2$.

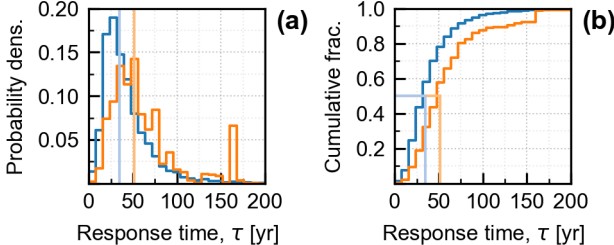

**Figure 6. (a) PDF and (b) CDF of estimated glacier response times ($\tau$). Blue lines are number-weighted and orange lines are area-weighted. Vertical dashed lines indicate the medians of each distribution. While there are many glaciers with short response times, they are generally small (either thin or spanning a large elevation range). Because of the longer response time of larger glaciers, over half of all Alaska glacier area we analyzed is contained in glaciers with response times greater than 50 years.**

In practice, glaciers with the shortest $\tau$ ($< 15$ yr) tend to be hanging glaciers and small valley glaciers, where steep slopes result in thinner ice and higher values of $b_t$. Examples include the smaller glaciers of College Fjord (Vassar Glacier, $\tau = 16$ yr; Holyoke Glacier, $\tau = 10$ yr), Cantwell Glacier ($\tau = 16$ yr), and Byron Glacier ($\tau = 15$ yr). However, small glaciers that widen below their ELA or terminate on shallow slopes (e.g., cirque glaciers) can occasionally have $\tau$ exceeding 50 yr (Laughton Glacier, $\tau = 57$ yr; Anderson Glacier, $\tau = 82$ yr; see also Barth et al., 2018). Glaciers with the longest response times ($> 100$ yr) tend to be large and also tend to terminate on shallow slopes. Some examples of glaciers with larger $\tau$ include Tweedsmuir Glacier ($\tau = 91$ yr), Tana Glacier ($\tau = 132$ yr), and Brady Glacier ($\tau = 140$ yr).

We note that $\tau$ has been estimated using the modern, rather than the preindustrial, glacier geometry. For most glaciers this is a small effect relative to the other uncertainties. The linear model closely emulates numerical flowline models through the course of several kilometers of terminus retreat (e.g., Christian et al., 2018) suggesting that the effective response time does not change rapidly with glacier state. The biggest errors are likely to be for the smallest glaciers that have undergone the largest fractional change in their geometry over the industrial era.

## 5 Estimating glacier disequilibrium

### 5.1 Disequilibrium for a linear warming trend

We first estimate glacier disequilibrium for a linear warming trend. We choose the year 1880 as the onset of the trend, consistent with the IPCC (Eyring et al., 2021) and related prior work on glaciers (Roe et al., 2017; Roe et al., 2021; Christian et al., 2022a). We analyze glacier response through 2020 (i.e., $t = 140$ yr in Eq. (4)). Figure 7 shows the PDF and CDF of $f_{eq}$. Weighted by number, the distribution has a median (and 90% range) of 0.57 (0.17, 0.83). Recall that a value of $f_{eq} = 0.57$ means the median glacier still has still has 43% of its total retreat ($L'_{eq}$) remaining to reach equilibrium with the present-day



climate (Eq. (5)). Weighted by area, the distribution has a median (and 90% range) of 0.41 (0.06,0.76). The opposing skew in the PDFs between the number-weighted and area-weighted distributions mirrors that of glacier number and area (Figs. 2b,c).

That is to say, smaller glaciers, of which there are many, tend to have small response times and are generally closer to equilibrium. Larger glaciers, which together constitute most of Alaska's glacierized area, are generally further from equilibrium, reflecting their larger response times (Table A1). For the largest category of glaciers we consider, the median $f_{eq} = 0.37$ implying, for these assumptions, that a committed retreat (i.e., $|L'_{eq} - L'|$) nearly twice the current retreat ($L'$) is already built into the glacier's future response.

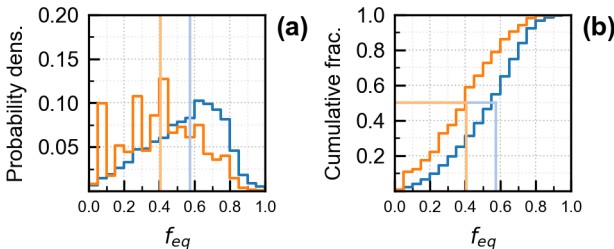


**Figure 7. (a) CDF and (b) PDF of estimated fractional equilibration ($f_{eq}$) for a 140-yr linear climate trend. Blue lines are number-weighted and orange lines are area-weighted. Lighter vertical lines indicate the medians of each distribution.**

**5.2 Shape of the warming trend**

Up to this point, we have approximated the industrial-era warming trend as linear so that $f_{eq}$ has an analytical solution (Eq.

(5)). In this section, we evaluate the degree to which the shape of the anthropogenic climate forcing over time can influence glacier disequilibrium. We consider two alternatives to the linear warming trend, which represent alternatives views on the shape of the anthropogenic warming.

The thin grey line in Fig. 8a shows Alaska-wide summertime (JJAS) temperature anomalies since 1880, taken from the

Berkeley Earth dataset (Rohde and Hausfather, 2020), and the thicker grey line applies a 30 yr low-pass filter (zero-phase second-order Butterworth). While there is substantial year-to-year variability, there has been an overall warming of ~1.2 °C since 1880. This is a typical value for summertime trends seen elsewhere on land (e.g., Allen et al., 2019). The blue line in Fig. 8a is the linear warming scenario, for which we presented results in Fig. 6. The orange line in Fig. 8a is the second warming scenario we consider in this study. It is characterized by a warming in the first half of the 20th century, a mid-century cooling,

and a resumed warming in recent decades. This scenario incorporates what is sometimes deemed the "*Early Twentieth-Century Warming*", or ETCW (Hegerl et al., 2018), which characterizes many, particularly high-latitude, temperature records. It is debated whether this shape is attributable to anthropogenic causes or reflects natural, internal climate variability (Haustein et al., 2019). The green line in Fig. 8a is the third warming scenario we consider. Its shape is an approximation of the Global Warming Index (GWI), which is an estimate of the global anthropogenic radiative forcing (Haustein et al., 2017). The change

in the rate of warming circa 1970 reflects an increase in the rate of industrial emissions. All three scenarios are standardized





to start at 0° C in 1880 and reach a value of 1.2° C in 2020. We note that $f_{eq}$ here depends only on the shape of the trend and not its magnitude. In this study, we consider only temperature trends. Observed annual precipitation trends in Alaska have generally been small over the 20th century (McAfee et al., 2013; Bieniek et al., 2014), and can be difficult to quantify across the whole region due to sparse early data (McAfee et al., 2014; Ballinger et al., 2023).


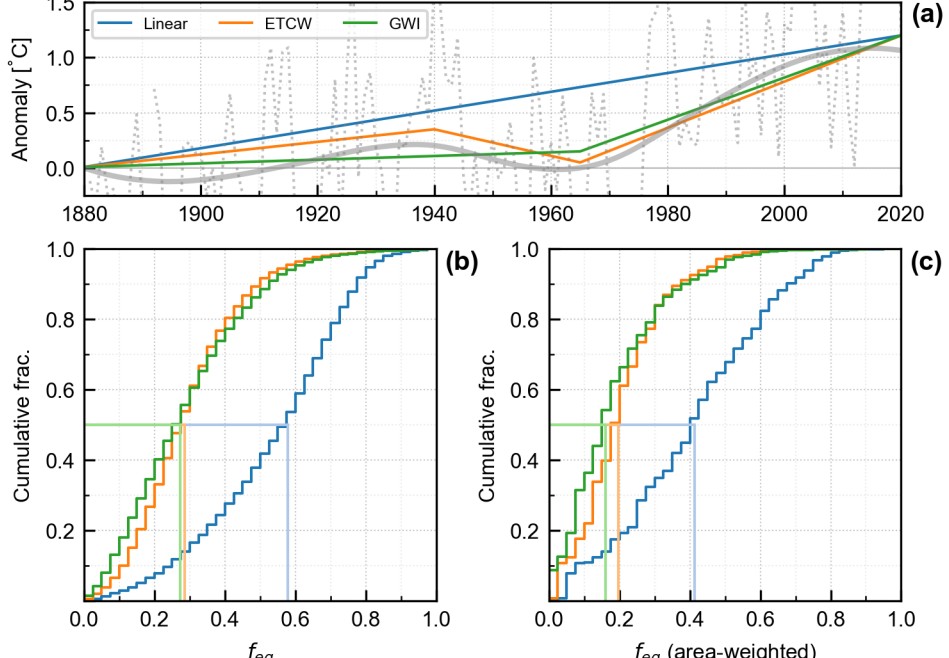

**Figure 8. Comparing glacier disequilibrium for three warming scenarios. (a) Observed annual Alaska-wide summertime temperature anomalies (JJAS) from the BE dataset (dotted grey), and after applying a 30 year low-pass filter (solid grey). Three anthropogenic warming scenarios are considered: linear (blue), ETCW (orange) and GWI (green). See text for details. Both observations and scenarios are set at 0 °C in 1880. The lower panels show (b) number-weighted and (c) area-weighted $f_{eq}$ calculated at 2020 for each warming trend. The blue lines are identical to those in Fig. 7b. Lighter vertical lines represent median values. The population medians and medians for different area categories defined by glacier area are reported in Table A1.**

To assess $f_{eq}$ for our three warming scenarios, we calculate $L'(t)$ using Eq. (3) discretized into time increments of one year, and our estimates of $\tau$ (Fig. 6). We assume that temperature anomalies, $T'(t)$, are related to mass balance by $b'(t) = \mu T'(t)$, where $\mu$ is a constant melt factor (units of m yr⁻¹ °C⁻¹), and we have $L'_{eq}(t) = \beta\tau\mu \cdot T'(t)$. Finally, we have $f_{eq} = L'/L'_{eq}$. Note that when the ratio of lengths is taken, both $\mu$ and $\beta$ cancel out, so our results do not depend on their values.

Figure 8 compares the effect of these three warming scenarios on $f_{eq}$ calculated for the year 2020. Relative to the linear scenario, both the ETCW and GWI scenarios show a greater degree of disequilibrium; their greater rate of warming in recent decades means that glaciers are further from equilibrium with the current climate, thus lowering $f_{eq}$. For the ETCW scenario,



weighted by number, the distribution of $f_{eq}$ has a median (and 90% range) of 0.29 (0.09, 0.59). Weighted by area, the distribution has a median (and 90% range) of 0.20 (0.03, 0.48). For the GWI scenario, weighted by number, the distribution has a median (and 90% range) of 0.27 (0.05, 0.62). Weighted by area, the distribution has a median (and 90% range) of 0.16 (0.02, 0.51).


Our results demonstrate that the shape of the warming has a substantial impact on $f_{eq}$, for both number- and area-weighted distributions. The median values of $f_{eq}$ are approximately halved in the ETCW and GWI scenarios compared to the linear scenario. Expressed another way: for the linear scenario, approximately 60% of glaciers have $f_{eq} > 0.5$ (i.e., are more than halfway equilibrated to the current climate); whereas for the ETCW and GWI scenarios, this falls to only 12% of glaciers (Fig.

8a). The differences among scenarios are somewhat less for the area-weighted distributions because there is more weighting given to the larger $\tau$ glaciers, for which the decadal-scale details of the warming matter less. It is interesting to note that the ETCW and GWI scenarios have similar $f_{eq}$ distributions (Figs. 8a,b). This is because we calculated $f_{eq}$ for the year 2020, for which the previous 50 years of climate history are similar in both scenarios (Fig. 8a). If we calculated $f_{eq}$ at 1940, for example, the distributions of $f_{eq}$ would be different for the two scenarios because of their divergent climate history over the prior

decades.

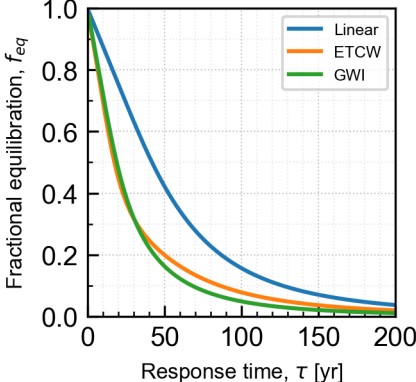

**Figure 9. Fractional equilibration in 2020, as a function of response time for the three warming scenarios shown in Fig. 8.**

**5.3 Sensitivity of the results**

The two factors that control $f_{eq}$ are $\tau$ and the shape of the anthropogenic-warming scenario, and both have some uncertainty. In order to assess the relative importance of these uncertainties, we can plot $f_{eq}$ (at 2020) as a function of $\tau$ and our three warming scenarios (Fig. 9). All three scenarios converge on the limits $f_{eq} \to 1$ for small response times ($\tau \to 0$ yrs) and $f_{eq} \to 0$ for large response times ($\tau \gtrsim 150$ yr). The largest difference between the linear and ETCW/GWI scenarios occurs for $\tau \sim 50$ yr, for which the differences in recent warming have the biggest impact. This $\tau$ is close to the area-weighted median we found



for our whole population ($\tau = 52$ yr; Table A1). Figure 9 can be used to evaluate how uncertainties in $\tau$ lead to uncertainties

in $f_{eq}$. The information is not available to perform a formal uncertainty analysis, so in previous work (Roe et al., 2017) we

assigned broad uncertainty to our estimates of $\tau$. We assumed a standard-deviation of $\tau/4$, meaning the 95% bounds (i.e.,

$\pm 2\sigma$) on the range for $\tau$ is equal to $\tau$ itself. In other words, the 95% bounds for $\tau$ spans a factor of three ($\tau/2$ to $3\tau/2$). We

consider this is a reasonable estimate for this study too. Applying this to Fig. 9 and the linear scenario, the largest uncertainty

in $f_{eq}$ arises for glaciers with $\tau \sim$ 20-60 yr. For such glaciers, uncertainty in $\tau$ and choosing a realistic warming scenario over

the linear trend have comparable importance. When choosing between realistic scenarios however, the uncertainty is small

compared to that of $\tau$. Note that, for most values of $\tau$, the symmetric uncertainty we asses around $\tau$ translates to asymmetric

uncertainty around $f_{eq}$. For glaciers with long response times (e.g., $\tau \gtrsim 150$ yr), uncertainties in $f_{eq}$ are low because $f_{eq} \sim 0$

across a wide range of $\tau$ regardless of warming scenario. For glaciers with small response times ($\tau \to 0$ yr), the absolute

uncertainty in $\tau$, and thus the uncertainty in $f_{eq}$, is lower than for the intermediate values of $\tau$, despite the steeper slopes of the

curves in Fig. 9 for smaller $\tau$. In summary, for any individual glacier, especially those with $\tau$ of a few decades, the value of

$f_{eq}$ can have large uncertainty because of uncertainty in both $\tau$ and the true shape of the warming trend.

We also tested alternative values of $H$ and $b_t$ in our estimation of $\tau$. First, we repeated our analyses using the thickness dataset

from Farinotti et al. (2019), which produces a lower median and narrower distribution of $H$ as compared to Millan et al. (2022).

Taking the GWI scenario as an example, using Farinotti et al. to estimate $H$ gives a number-weighted median (and 90% range)

for $f_{eq}$ of 0.35 (0.10, 0.72), compared to 0.27 (0.05, 0.62) for Millan et al. (2022). When weighted by area however, the

difference in the resulting median and 90% range between datasets is negligible. This contrast between weighting methods is

mainly attributable to the largest glaciers having a greater estimated thickness in the Farinotti et al. dataset.


Secondly, we tested the impact of altering the AAR in our calculation of $b_t$. Zeller et al. (2023) provide AARs for a large

population of Alaskan glaciers based on satellite imagery of end-of-summer snow cover. They report an average AAR of 0.4

for all glacier area in Alaska over the period 2018-2022. Using this dataset, we examine two variations on our analysis: i)

applying the average AAR of 0.4 uniformly (AAR$_{0.4}$) to match our standard analysis (AAR$_{0.6}$), and ii) using the AARs Zeller

et al. derived for individual glaciers (AAR$_Z$). The resulting values of $b_t$, $\tau$, and $f_{eq}$ are summarized in Table D1, using the GWI

warming scenario as an example.

In our model, a smaller AAR yields a higher ELA, resulting in more negative $b_t$ and thus a smaller $\tau$. Relative to the standard

analysis, the AAR$_{0.4}$ case corresponds to a number-weighted (and area-weighted) increase in ELA of 100 (250) m (not shown).

Despite sharing a median AAR, the number-weighted (and area-weighted) median ELA is an additional 150 (20) m higher in

the AAR$_Z$ case than in the AAR$_{0.4}$ case, suggesting individual AARs may be particularly relevant for representing small

glaciers. For our population, using individual AARs yields significantly more negative estimates of $b_t$ than assuming an



equivalent uniform AAR. The size of this effect is comparable to the difference between the AAR$_{0.4}$ case and the standard
AAR$_{0.6}$ case. When binned by glacier area (as in Table A1), the difference in $b_t$ among cases is notably consistent in magnitude.

However, differences among cases are proportionally larger for glaciers with less negative $b_t$, which typically also have smaller
$H$, corresponding to greater sensitivity in small values of $\tau$. Of glaciers with large $\tau$, sensitivity decreases with more negative
$b_t$ for the same reason. Figure 9 shows that sensitivity in $f_{eq}$ is amplified for small $\tau$, and damped for large $\tau$. Because large
glaciers have long response times, we find that our estimates of area-weighted $f_{eq}$ are more insensitive to our estimated AAR.
Overall, these sensitivity studies show our analyses remain consistent to alternative dataset choices and can readily be updated

as new datasets and better observations are obtained.

## 6 Summary & Discussion

In this study we have estimated the state of disequilibrium for a large population of Alaskan glaciers. We excluded small
glaciers and tidewater glaciers because our model of glacier dynamics is less applicable to such systems, leaving us with a
population of approximately 5600 glaciers representing 79% of the region's glacier area. There are considerable observational

uncertainties in glacier thickness and mass balance, which means that estimating disequilibrium for any single glacier involves
substantial uncertainty. By analysing a whole population of glaciers, we aim to provide a region-wide picture, without
depending on the uncertain details of one particular setting.

Building on previous work, (Christian et al., 2022a) we define a metric of disequilibrium that uses a simple linear model of ice

dynamics; and uses characteristic ice thickness, $H$, and terminus mass balance, $b_t$, to estimate the glacier response time, $\tau$. A
key advantage of our metric is that, because it is a ratio, other geometric parameters cancel out. One disadvantage of using our
linear model is that parameters are fixed, so the predicted retreat does not account for settings where the glacier geometry (i.e.,
width, bed slope) varies strongly upslope of the current margin. More sophisticated and detailed numerical modeling might
address such settings. However, given the substantial uncertainties in input parameters and initial conditions, and given our

focus on region-wide disequilibrium, we do not think that a more complex modelling system would necessarily provide better
assessments. Ice thickness and mass-balance gradients still must be estimated in the absence of direct observations, and biases
in these estimates will still propagate into the results of a model with more sophisticated ice dynamics.

We provide results as statistical distributions for our selected population of glaciers, both number weighted (i.e., by count),

and area weighted. Table A1 presents our results in separate categories defined by glacier area. Our estimated distributions of
$H$, $b_t$, and $\tau$ appear reasonable relative to published estimates for individual glaciers. We defined fractional equilibration, $f_{eq}$,
as the ratio of current retreat to the eventual equilibrium retreat if the climate were to immediately stop changing. Our analyses
identify two key factors affecting $f_{eq}$: glacier response time and the shape of the anthropogenic forcing, and we also
demonstrate that there is interplay between them. Small-$\tau$ glaciers ($\tau \rightarrow 0$ yr) are in near-equilibrium with the current climate





($f_{eq} \rightarrow 1$), and so do not depend sensitively on climate history. Large-$\tau$ glaciers ($\tau \gtrsim 100$ yr) are far from equilibrium ($f_{eq} \rightarrow$ 0), and so respond slowly enough that decadal-scale details of past climate trajectory also do not matter. However, for intermediate-$\tau$ glaciers (representing the majority of glaciers in our population), $f_{eq}$ varies sensitively and depends on both the shape of the forcing and on $\tau$.

The linear warming trend has an analytic solution for $f_{eq}$, which is attractive as a first approximation and provides insight into the physical factors affecting how $f_{eq}$ evolves over time. However, it is likely that the ETCW and GWI scenarios are better representations of the true anthropogenic forcing, both of which imply that glaciers are further from equilibrium than in the linear scenario. These two scenarios have similar-enough climate histories over the past five decades that the values of $f_{eq}$ are also similar (Figs. 8,9). Taking just the GWI scenario, when weighted by number, we estimate the distribution of $f_{eq}$ has a

median (and 90% range) of 0.27 (0.05, 0.62). Weighted by area, the distribution has a median (and 90% range) of 0.16 (0.02, 0.51). These results indicate that, overall, Alaskan glaciers are in a state of dramatic disequilibrium due to anthropogenic climate change, with substantial continued retreat already guaranteed.

For hydrologic-resource planning, or for other local purposes, the state of disequilibrium of one specific glacier may be of

importance. Our algorithm is useful for providing an initial estimate, but it should be considered only as a rough approximation. A more comprehensive assessment might be made for individual glaciers, using more of the available information. For instance, if the vertical profile of the mass balance is known then, for a given catchment geometry, the glacier extent that would be in equilibrium with that profile can be estimated (e.g., Rasmussen and Conway, 2001). One challenge is that mass balance remains poorly observed in most locations. Moreover, due to large interannual variability, it can take many years of

observations to accurately assess the long-term position of the ELA, which is all the harder in the face of long-term trends. For specific locations, it is often possible to assess the accuracy of the glacier-thickness estimates using radar measurements. Given the severe disequilibrium indicated in our results, it seems important to assess the impact of varying thickness on glacier simulations. For individual glaciers, it may be that more complex ice-dynamics modelling (2D or 3D) over the realistic catchment geometry offers improved estimates of disequilibrium, although that should always be evaluated relative to the other

uncertainties in the setting. Finally, there is an increasing availability and use of remotely sensed imagery (e.g., Maurer et al., 2019; Shean et al., 2020; Friedl et al., 2021; Jakob and Gourmelin, 2023; Knuth et al., 2023). In the face of warming, glacier adjustment takes the form of overlapping sequential stages of i) thinning, ii) reduced fluxes, and iii) retreat (e.g., RB14), all of which can now be observed with remote sensing. It may be such observations can be combined to estimate the dynamical phase of a retreat.


Our algorithm offers a straightforward and efficient method to assess glacier disequilibrium at regional scales. One robust result is that the wide range of glacier response times implies a broad distribution in the degree glacier disequilibrium within





any region. Constraining current disequilibrium is important for the numerical modelling of glacier populations. In particular, care must be taken to properly represent the initial conditions of glacier simulations. It should also be recognized that
uncertainty in climate history and glacier response time means significant uncertainty is attendant on disequilibrium estimates (especially for individual glaciers), and likely points to the need for ensemble modelling techniques to provide probabilistic projections of future glacier evolution (e.g., Christian et al., 2022b). Finally, our assessment of Alaskan glaciers implies a severe degree of glacier disequilibrium with the current climate. It seems important for policy makers, resource managers, and the general public to appreciate how much retreat is already baked into the future evolution of glaciers.

**Acknowledgements**

D.R.O and G.H.R acknowledge support from NSF grants: AGS2102829 and GLD2314212, and thank Louis Sass for valuable feedback.

**Appendix A**

Table A1 shows summary statistics for our population of glaciers categorized into bins by area. The median value within each
bin is reported along with the 90% range. The median values for the magnitudes of $H$, $b_t$, and $\tau$ increase with glacier area. However, the 90% ranges are wide and often overlap the ranges of adjacent area categories, reflecting the wide range of individual glacier geometries in each. As shown by Fig. 1b,c, there are fewer glaciers with large areas. The geometry for these few largest glaciers is much more influential to our area-weighted estimates, highlighting their individual characterization as a key target for improving estimates of disequilibrium.


| Glacier area [km$^2$] (90% range) | Count | Bin Area Frac. | Area [km$^2$] | $H$ [m] | $b_t$ [m yr$^{-1}$] | $\tau$ [yr] | $f_{eq}$ (linear) | $f_{eq}$ (ETCW) | $f_{eq}$ (GWI) |
|---|---|---|---|---|---|---|---|---|---|
| (1, 5) | 4017 | 0.13 | 1.8 (1.0, 4.2) | 73 (40, 129) | -2.1 (-4.3, -0.9) | 34 (13, 96) | 0.59 (0.17, 0.84) | 0.29 (0.09, 0.61) | 0.28 (0.05, 0.64) |
| (5, 25) | 811 | 0.13 | 8.4 (5.2, 21) | 129 (71, 231) | -3.7 (-6.8, -1.9) | 33 (16, 82) | 0.60 (0.22, 0.80) | 0.30 (0.11, 0.54) | 0.29 (0.07, 0.57) |
| (25, 100) | 240 | 0.18 | 42 (26, 91) | 225 (134, 379) | -5.3 (-8.5, -2.9) | 44 (21, 99) | 0.48 (0.16, 0.74) | 0.23 (0.08, 0.44) | 0.20 (0.05, 0.46) |
| (100, 250) | 69 | 0.16 | 149 (104, 238) | 367 (228, 484) | -6.7 (-10.4, -3.6) | 51 (29, 102) | 0.42 (0.15, 0.65) | 0.20 (0.08, 0.34) | 0.17 (0.05, 0.34) |
| (250, 3363) | 42 | 0.40 | 447 (293, 1171) | 427 (229, 640) | -7.3 (-10.8, -2.7) | 57 (31, 146) | 0.37 (0.08, 0.63) | 0.18 (0.04, 0.32) | 0.14 (0.02, 0.32) |
| **Total** | 5179 | 1.0 | 2.2 (1.1, 37) | 81 (42, 230) | -2.4 (-6.0, -1.0) | 35 (14, 96) | 0.58 (0.17, 0.83) | 0.29 (0.09, 0.59) | 0.27 (0.05, 0.62) |
| | | | | | | | | | |
| **Total (by area)** | 5179 | | 158 (2, 3191) | 331 (70, 644) | -5.7 (-10.2, -1.9) | 52 (19, 167) | 0.41 (0.06, 0.77) | 0.20 (0.03, 0.48) | 0.16 (0.02, 0.51) |



**Table A1. Median calculated values binned by glacier area. Parentheses are the 90% range. Note that, due to the shape of the distribution of glacier areas in Fig. 1b, the areas of glaciers within each bin are skewed towards the lower bound. The summary statistics for the population are shown below. These values are equivalent to the number-weighted statistics reported in the main text. For comparison, area-weighted statistics for the population are included in the bottom row.**

**Appendix B**

Table B1 presents the Alaska mass-balance data we compiled to produce a representative region-wide value for $db/dz$ in the ablation zone (i.e., Fig. 4). The estimated $db/dz$ values for each glacier and a summary of the data evaluated is shown in Table B1. Because there are few glaciers with long-term records, we include shorter records where there are sufficient measurements in the ablation zone. For glaciers with long records, we find the shape of the balance profile and magnitude of $db/dz$ to be
similar among different years.

| | $db/dz$ [m yr$^{-1}$ km$^{-1}$] | Std. err. [m km$^{-1}$] | N years | N obs. | Observation years | Observation $dz$ [m] | Glacier $dz$ [m] | Source |
|---|---|---|---|---|---|---|---|---|
| Black Rapids Glacier | 6.4 | 0.15 | 45 | 235 | 1972-2016 | 900-2230 (1330) | 720-3080 (2360) | WGMS, 2023 |
| Columbia Glacier | 6.4 | 0.57 | 3 | 38 | 1978-2011 | 140-1460 (1320) | 0-3690 (3690) | WGMS, 2023 |
| East Yakutat Glacier | 6.5 | 0.54 | 3 | 47 | 2009-2011 | 110-680 (570) | 30-1670 (1640) | WGMS, 2023 |
| Eklutna Glacier | 6.9 | 0.69 | 8 | 35 | 2008-2015 | 1120-1540 (420) | 540-2050 (1510) | Sass et al., 2017 |
| Exit Glacier | 7.8 | 1.1 | 7 | 32 | 2011-2017 | 870-1290 (420) | 120-1600 (1480) | Kurtz, personal communication, 2022 |
| Gilkey Glacier | 6.4 | 0.69 | 3 | 6 | 2012-2014 | 240-1500 (1260) | 110-2360 (2250) | Young, personal communication, 2022 |
| Gulkana Glacier | 7.8 | 0.25 | 54 | 212 | 1966-2019 | 1260-2030 (770) | 1160-2440 (1280) | USGS, 2016 |
| Kahiltna Glacier | 3.3 | 0.70 | 1 | 9 | 2011-2011 | 790-1400 (610) | 260-5130 (4870) | Young et al., 2018 |
| Lemon Creek Glacier | 10.5 | 1.3 | 6 | 40 | 2014-2019 | 820-1240 (420) | 680-1490 (820) | USGS, 2016 |
| Mendenhall Glacier | 9.9 | 0.43 | 11 | 52 | 1997-2010 | 100-1570 (1470) | 90-1970 (1880) | Boyce et al., 2008; Young, 2022, personal communication |
| Wolverine Glacier | 7.5 | 0.28 | 54 | 191 | 1966-2019 | 550-1370 (820) | 430-1640 (1210) | USGS, 2016 |

**Table B1. Summary of mass balance data evaluated for Fig. 4. The first and second columns give the slope ($db/dz$) and standard error of the least-squares fit of annual point mass balance on elevation. Mass balance data were detrended for time. The remaining**





**columns show: the range of years where mass balance observations were evaluated, the total number of point observations in each regression, the range of elevations among observations, and the glacier's total elevation range for reference.**

**Appendix C**

For valley glacier geometries, intuition suggests that a glacier with greater area is typically also thicker and able to sustain a terminus extent with greater ablation. These tendencies of $H$ and $b_t$ offset each other in our distribution of estimated $\tau$ and result in a smaller spread than would be expected for independent variables (which can be seen from the spreads of $H$, $b_t$, and $\tau$ in the different size categories in Table A1). $H$ and $b_t$ are indeed somewhat anticorrelated for glaciers in our population ($r$ = -

0.52). We find a geometric relationship between $\tau$ and glacier area in our population for glaciers larger than 5 km². For a linear regression of $\log(\tau)$ on $\log(\text{Area})$, every doubling of area corresponds to a predicted $\tau$ that increases by a factor of 1.15. Although it only explains 9% of the variance in log-log space, the large number of data points mean the relationship is highly robust (Fig. 9). When using the Farinotti et al. (2019) thickness dataset to estimate $\tau$, we find a larger scaling factor of 1.22, with the relationship explaining 22% of the variance (not shown). For $\tau$ derived from the Farinotti et al. dataset, glaciers smaller

than 5 km² show a weak positive correlation with area, yielding a scaling factor of 1.08. The relationship explains less than 1% of the variance but is statistically significant and distinct from that of larger glaciers. No corresponding relationship is observed when using the Millan et al. (2022) dataset.

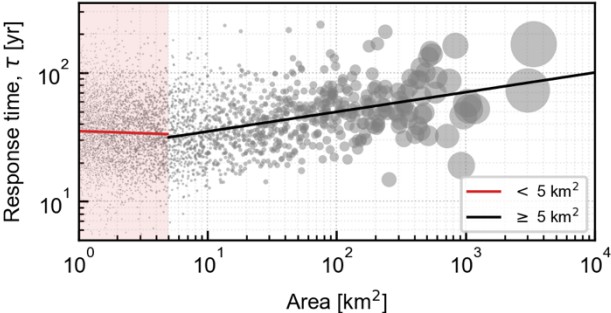

**Figure C1. Log-log relationship between glacier area and response time, $\tau$ using the Millan et al. (2022) dataset. Lines show the least-**
**squares linear fit to the log-transformed data for glaciers with area < 5 km² (red), and ≥ 5 km² (black). The black line has the equation: $\tau = 25\ yr \times (A/A_0)^{0.15}$, where $A_0 = 1\ km^2$. A doubling in glacier area is associated with $\tau$ increasing by a factor of 1.15 (SE = 0.014; $R^2 = 0.09$). The red line shows the linear fit for glaciers with area < 5 km², where there is no equivalent correlation. Points are scaled by area for visual clarity.**

**Appendix D**

Table D1 shows summary statistics from the alternative AAR cases we examine in Sect. 5.3, using the GWI warming scenario as an example. The results from the standard analysis are included for comparison, denoted as the AAR$_{0.6}$ case here. The median and 90% range of the affected variables are reported for each case, weighted both by number and by area. The



difference between the $AAR_{0.6}$ and $AAR_{0.4}$ cases represents the sensitivity to the AAR value chosen. The difference between the $AAR_{0.4}$ and $AAR_Z$ cases demonstrates the impact of assuming a single AAR for all glaciers.

| | $b_t$ [m yr$^{-1}$] | | $\tau$ [yr] | | $f_{eq}$ (GWI) | |
|---|---|---|---|---|---|---|
| Weighting | Number | Area | Number | Area | Number | Area |
| $AAR_{0.6}$ | -2.4 (-6.0, -1.0) | -5.7 (-10.2, -1.9) | 35 (14, 96) | 52 (19, 167) | 0.27 (0.05, 0.62) | 0.16 (0.02, 0.51) |
| $AAR_{0.4}$ | -3.1 (-7.5, -1.4) | -7.9 (-12.7, -2.5) | 27 (11, 71) | 41 (15, 85) | 0.37 (0.09, 0.70) | 0.22 (0.07, 0.59) |
| $AAR_Z$ | -4.1 (-9.0, -1.8) | -8.7 (-12.1, -3.4) | 21 (8, 55) | 38 (13, 74) | 0.47 (0.14, 0.77) | 0.25 (0.09, 0.64) |

**Table D1. Median calculated values for versions of our analysis using alternative AARs. Parentheses are the 90% range. Values for each variable are given weighted both by number and by area. Results from the standard analysis, denoted as the $AAR_{0.6}$ case, are included for comparison with the two alternative cases. The $AAR_{0.6}$ case is identical to the main analysis. The $AAR_{0.4}$ case applies a uniform AAR of 0.4 for all glaciers. The $AAR_Z$ cases uses individual glacier AARs provided by Zeller et al. (2023) based on satellite imagery of end-of-summer snow cover.**

**Code and data availability**

The code to produce data and figures are available at https://doi.org/10.5281/zenodo.13968460.

**Author contribution**

The analysis was conceptualized by DO, GR, and JEC, and performed by DO. DO and GR prepared the manuscript with contributions from JEC.

**Competing interests**

The authors declare that they have no conflict of interest.

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
