# Peer review of "An assessment of the disequilibrium of Alaska glaciers"

_EGUsphere, 2024_

## Referee Comment (RC1)

In this study the authors use a simple model of glacier dynamics, based on mass continuity and assumptions about spatial variations in mass balance, to assess the disequilibrium of Alaska Glaciers. The advantage to this approach, as opposed to using higher order modeling, is that it can be done quickly for large numbers of glaciers without requiring a lot of inputs. The authors conclude that Alaska glaciers are in a severe state of disequilibrium, having only undergone 27% of the retreat necessary to be in equilibrium with today's climate. The observation that the glaciers are in a severe state of disequilibrium is perhaps not too surprising, and the simplicity of the model and uncertainty of the data used in the model raises some questions about exactly how far the glaciers are from equilibrium. To me, the more interesting result is the discussion of how the climate warming scenario affects glacier disequilibrium (the increased warming trend over the past several decades is much more important than the warming that occurred in the early 20[th] century) and how glacier geometry affects disequilibrium. I found myself wondering whether the authors have tried looking at spatial variations in glacier disequilibrium. I think it could be interesting to see whether glaciers in certain parts of Alaska are closer/farther from equilibrium.

Overall I think this is an interesting and thought provoking paper. I have a few general comments on the paper, mostly aimed at clarifying the glacier model and assumptions associated with it, as well as handful of specific comments.

**General comments**

1. Section 2 is important for setting the stage for the rest of the paper and laying out the model assumptions. With that in mind, I think this section needs some additional details and clarifications, and perhaps some re-structuring.
   a) The section immediately starts off with a figure illustrating the idea of committed retreat, but the figure is based on equations that appear much later in the section. Consider starting with the model description.
   b) I'm not super familiar with the linearized model, and so in reviewing this paper I spent some time also reading Roe and Baker (2014) and Christian et al. (2018), as well as Roe and O'Neal (2009). It appears to me that the model really originates in Roe and O'Neal, or at least is mostly explicitly spelled out in the appendix of that paper. Consider citing that paper.
   c) I think the model assumptions need to be spelled out more clearly. As I understand it, the linearized model assumes (i) an initial steady-state climate and glacier geometry and (ii) small departures from steady-state. I think the authors are attempting to address (i) in lines 65-69, but not very explicitly and it can be easy to miss this assumption because it is presented a few paragraphs before the model.
   d) Equation 2 is described as the equilibrium length response to a step change in climate; isn't it just a change in a climate from one steady-state to another? Then, equations 3 and 4 describe the length anomaly and fractional equilibration for linear climate trends, but the sentence after equation 4 says that it applies for any climate trend. Can these equations all just be written in terms of a general change in climate? It's a bit confusing as is.

2. The fractional equilibration (equation 4) depends on the time since the start of the climate trend, the glacier thickness, and the mass balance at the terminus. The authors use $t_0=1880$, ice thickness from regional glacier thickness maps that are known to have large errors, and mass balance profiles from (presumably) representative glaciers. The authors do test the sensitivity of their results to some extent, but I also wonder if it would be useful to use propagation of uncertainty to explicitly show how the model results are impacted by the uncertainty in the model parameters. And I may have missed it, but I don't think the authors tested the impact of their choice of $t_0$ other than to reference another study.

3. The captions for Figures 3-7 indicate that the panels show the probability density functions and cumulative distribution functions (but note that the caption for Fig. 6 seems to have these flipped). However, I don't think that is the correct terminology. These look like histograms. When you integrate a PDF from $x_1$ to $x_2$, you should get the probability that a sample falls within that range, whereas here you are showing the fraction of the samples that fall within predefined bins. Personally I think the better way to plot this data would be use to empirical cumulative distribution functions and complementary distribution functions. I think histograms can sometimes be misleading when sample sizes are small, such as at the tails of the distributions. Regardless of how you choose to plot the data, make sure that the terminology is correct.

**Specific comments**

Title and throughout: This is pedantic, but I think it should be "Alaska glaciers" and not "Alaskan glaciers" since "Alaskan" is an adjective. For example, see
https://www.adfg.alaska.gov/static/home/library/pdfs/writersguide_section6.pdf

L12-13 and later: I'm struggling to wrap my head around the area weighting. I under the equations, but it's not entirely clear to me what the weighting does.

L28: Do I understand correctly that the Johannesson time scale emerges naturally in your model, which is why it makes sense to use it in this analysis instead of that of Harrison (for example)? Might be good to point that out.

L42: Models, at least if done correctly, should be spun up with the climate so that they are capturing the current dynamical state.

L46: The response time depends on geometry, so I'm not sure why you are also mentioning geometry here.

L60: How can they be identical if their response times are different? They must have different geometries and/or climates.

L62: "… change in climate from a steady state." ?

L68: Suggest "will tend to recover on its own" or "will recover on its own if there is no trend in climate".

L180: Is it okay to treat \tau as a constant and not something that changes during a warming trend? Maybe something to discuss in Section 2. And also, here you state that you are estimating \tau prior to the start of the warming trend but you are using modern thickness estimates. (I see that you come back to this later in line 271, but again, I think being more upfront about the model and assumptions would be helpful.)

L280: It took me a little while to remember that f_{eq} was fractional equilibration and to remember what that meant. It could be helpful to remind the reader occasionally; for example, in Figure 7 you could consider indicating that the left side of x-axes indicates more committed retreat and the right side indicates less committed retreat.

L295: I guess this should be Eq. (4)?

L323-326: This gets back to my confusion about the model that I discuss in the general comments. Are you using a linear trend for each year? Meaning that Equation 4 is for a linear trend, and so my confusion is related to lines 110-111. So it seems like you are essentially applying a perturbation each year. Are you also then updating the ice thickness and response time? A glacier's response time will change quite a bit as it thins and retreats.

L431: Do we know how this "dramatic disequilibrium" compares to past climate events?

---

## Referee Comment (RC2)

**Review of An assessment of the disequilibrium of Alaskan Glaciers**
**By Otto et al.**

SUMMARY

This study investigates the state of disequlibrium of ~5600 glaciers in Alaska using a well-published "model"/equation that links mass-balance anomalies and retreat in terms of length. The analyses focus on how different response times and the shape of the climate forcing affect the state of disequilibrium. Overall, they find the glaciers are in a state of severe disequilibrium and thus a lot of retreat is already inevitable in the future.

Overall, the study is highly relevant and provides new knowledge about the state of Alaskan glaciers, thereby making it suitable for The Cryosphere. However, there are considerable areas of the manuscript that could be revised to improve readability and more clearly convey these findings. Specifically, the method descriptions were fairly hard to follow, even though the concept itself is fairly "straightforward" (i.e., there's a model that can determine the state of disequilibrium and this is stated with the f_eq metric). In fact, the "straightforwardness" of the analysis is one of the most eloquent pieces of the analysis; hence, if the methods and readability can be improved, I think it will reach a much larger audience and the key points will come across much better. I thus recommend major revisions.

MAJOR COMMENTS

Section 2 is a bit hard to read, which seems to be due to the structure of the section. For example, the first sentence refers to Figure 1, which ultimately took me to read Figure 1; however, without any context, it was very hard to understand. Another issue with this section is that Figure 1 is technically a result; however, there's no information about the model and methods given yet – that all comes later. It thus feels very confusing and forced me to read it twice in order to understand, i.e., understand the methods first, and then understand this theoretical result. It also seems to provide remarkably detailed justification related to not focusing on natural interannual climate variability without any context. If you want to keep this theoretical case (i.e., results) as a way of describing the concept within this section, I would at a minimum recommend that you restructure the section so that the text is logical to follow in a single read. I would further recommend that you focus on the key elements, and provide detailed justification as relevant later on.

Section 2 and 3 – check on formatting requirements, but it's a bit odd to see equations that are written in both equations as well as equations written in lines (e.g., L96, 97, 105, L158, etc.).

The heavy use of variables in place of actual text reduces readability. Even something as simple as writing out tau as "glacier's response time" or just "response time" or $H$ as characteristic thickness could greatly improve accessibility and readability of the manuscript and only add one word. This is also likely part of the reason Section 2 is challenging to read.

Can Figures 3, 5, 6, 7 be merged together? This would be nice to able to see the differences. Also, it's interesting that the regional data is provided in Figure 2; however, these regional differences aren't really included within the analysis. It could be interesting to see Figures 3, 5,

6, 7 with these regional colors included within the pdfs. I recognize this would not enable the area-weighted and count pdfs to be done, so there's some discretion as to the important narrative to show. Maybe there could be a Figure 3, 5, 6, 7 pdf with just the counts in the supplement/appendix?

There are many parts where it seems like results are provided in the methods and methods are provided in the results. See specific comments below, but these really break up the flow of the text.

Discussion: I was surprised not to see any comparison to prior values of the state of disequilibrium or committed mass losses? It seems like referencing these studies, especially those in Alaska (e.g., Davies et al. 2024, *Nature Communications*) is needed at a minimum. I suppose that this model can only do length changes and not mass change. This may be an important distinction that should be highlighted in the discussion?

The glacier-wide climatic mass balance is the response of the glacier to the climate. The equilibrium state is thus the geometry that is needed to support a given climatic mass balance gradient.

LINE COMMENTS

L32 – consider replacing the "-" with "as" to make it easier to read.

L42 – This sentence feels out of place as it's a result but stated as a "for example" in the introduction. I don't think it's needed and distracts more than helps.

L55-56 – consider rephrasing. The "…, and for three different assumed …" doesn't make sense.

L70 – this sentence describing the equilibrium length, if I understand this correctly, then this is the length if you were to hold the temperature anomaly at any given point and allow the model to run until it reaches it's equilibrium length? If so, some mention of the timescales being different (i.e., it's a seemingly infinite timescale) is important to note here; otherwise, it looks like the glacier retreats that length immediately which does not make sense.

L70/71 – length anomaly should specify what the anomaly is calculated with respect to, i.e., the initial length.

L71 – consider replacing the ":" with "as" to make it easier to read.

L74 – why use years at all? The use of years implies it's associated with a specific time period. If it's intentional, i.e., that it reflects the change in temperature from pre-industrial, then this should be stated along with the justification. At the same time, I suspect that this would become exceptionally tricky here given Arctic amplification; hence, you have variable increasing temperatures for different areas, so the justification should make it very clear (and explicit) as to how it's being selected.

L78 – I don't know if this level of detail is necessary as the concept is fairly straightforward, but for my own curiosity, what happens for overshoot scenarios? I assume the current text is assuming only scenarios of consistently increasing temperatures?

L80 – regarding "increase their rate of change until it matches that of the warming trend", how is that possible given they're completely different units (e.g., m/yr versus degC/yr)?

L80 – "and so asymptote to as" doesn't make sense. This reads as though asymptote is a verb. If I understand what's being implied it's referring only to the orange line? The current sentence reads quite general so I was very confused since the 75-year response time example certainly has not reached its new equilibrium in this example.

L80 – this also assumes that the glacier is large enough, i.e., there's enough mass loss (or length change remaining) to enable this.

L81 – choose a different word than "spun up" as this implies initialization conditions of a model, which is not the case here.

L90 – "circumventing uncertainty in some inputs" is clearly model specific. I would state this as other readers may be interested in applying these metrics to other models (for example) and thus this might not apply. Note: see major comment as we still don't have any knowledge of "some inputs" yet, so this is also confusing to read.

L102 – can you add the context as to which metric (length, time scale, mass) this "estimate" refers to?

L106 – "… within 5% of the asymptotic limit (e.g., Fig 1c, orange line)" – this is wonderfully clear! This type of statement following the description of the method is very valuable and I'd encourage it to be earlier.

L107-108 – This reads as an important result, but in a conceptual/methods section. Suggest moving to the results section.

L109 – is this different than the "climate trend" described by L105? If different, specify this.

Equation 4 – why is there a box around this?

Figure 1 – can you show the initial length on one of these figures? It'd be nice to see the raw metric as opposed to only the anomalies/metrics.

L118 – "retreat" not "retreats", no?

L119-120 – again, it would be useful to show the metrics (and potentially state the differences) instead of only stating the differences here.

L137 – What about debris-covered glaciers? I assume their retreat behavior would complicate this?

L139 – This would read better deleting or changing "Lastly" since it refers to the last two.

Figure 2 caption – I assume the color-coding in Figure 1b and 1c refers to the subregions. It'd be good to explicitly state this.

Figure 2 – Why not use something different besides circles for the long-term mass balance records to avoid confusion?

Figure 2 – I assume you're aggregating these areas based on the center lat/lon? It looks like a regular grid of some sort. Would be good to state the resolution of this grid.

L154-157 – perhaps good reason to add a CDF (even if it's just the total) to these two plots as well as a right-hand side axis.

L157 – delete "simple". It's a glacier count. This implies glacier area is complex somehow?

Section 4.1 – Shouldn't this be "Characteristic thickness"?

Figure 3 – change "the method outlined in the main text". If you want to highlight this, state (see Section X) or something similar.

L209 – I assume you're referring to Order 2 RGI regions? If so, state this.

Figure 4 – Is there a reason that you include the accumulation areas (i.e., > 0 m/yr) when the methods state you're only using the ablation area data? Does it change the numbers you get at all?

L233 – no need to repeat the caption. Recommend re-writing such that you just state what the b_t values are and then refer to Figure 5a,b in parentheses.

L250 – Figure 6 shows … is just repeating the caption.

L251 and others – I don't think you need to state (and 90% range) every time. Only at first use.

L264 – can you give the areas or just a broad definition here of how "small" is used? < 10 km2 for example?

L299-314 – this is all methods.

Figure 8 – suggest using different colors to avoid confusion with the same colors being area-weighted vs. count in Figures 3, 5, etc.

L319 – remove "see text for details" as that's obvious. Refer to a section if essential.

L329 – Recommend not using highly unique acronyms like "ETCW" and "GWI" as they greatly impede readability. If writing three to four words is deemed too large, then come up with a new name to refer to them as that's meaningful.

L336 – The shape of the warming is important compared to the linear shape; however, the "ETCW" and "GWI" shapes provide relatively consistent values. This seems worth mentioning earlier (noted it's in L341-342) since it currently reads that they're all highly different until you get there.

L336 – Building on the impact of the shape, I would recommend analyzing this relative to the linear $f\_eq$, i.e., reporting the differences on a per glacier basis. This would make it a lot easier to grasp the statements like L338-340. L330-334 report the 90% range, which is highlighting the glacier variability, but ultimately this whole paragraph is about the differences on a per-glacier scale.

L343-345 – I don't see where this statement is supported?

L349 – check formatting.

Figure 9 – This looks highly smooth. I assume some function was fit through the data? Please state this in the caption.

L356 – what information is not available? This is quite vague.

L356-359 – is this not methods?

L350-367 – prior to this section, there are a lot of values provided. Then in this section, there are no values to actually accompany the uncertainty estimates. Instead the uncertainty is just qualitatively described and not shown. Can one provide some values? Maybe for some of these different examples?

369-375 – Like my prior comment, the interesting thing here is the difference by glacier, not the actual values. Consider changing the way this is provided. For example, are all glaciers 0.08 higher or is it highly variable? Is there not a figure that shows this that can be referenced as well?

L380 – suggest pulling out a few important numbers from the table here to make this more quantitative. Again, it'd be interesting to know the differences, not the values for all of them (see prior comments).

L389-392 – Are these statements supported anywhere? Figure 9 does not seem to show this.

L406-411 – Consider rephrasing. It's a bit odd to state that more detailed numerical modeling can address a problem but then state that you don't expect it to be any different. In that case, there's not really a point in having a more detailed numerical model.

L445 – "other uncertainties in the setting" – provide some examples. Otherwise, highly vague.

L458 – "It seems important" consider making this line stronger. Awareness of how much retreat is already baked into the future evolution of glaciers is important for policy makers, resource managers, and the general public to know to _____. Perhaps that _____ is preserve the glaciers that remain? Reduce our greenhouse gas emissions?

---

## Author Comment (AC2)

Green = reviewer comments, black = responses

We thank the reviewer for their careful reading and thoughtful comments. Their perspective has helped to significantly improve the clarity of the manuscript and we are grateful for their time spent evaluating our work. We have responded with comments interspersed into their review.

In this study the authors use a simple model of glacier dynamics, based on mass continuity and  assumptions about spatial variations in mass balance, to assess the disequilibrium of Alaska  Glaciers. The advantage to this approach, as opposed to using higher order modeling, is that it can  be done quickly for large numbers of glaciers without requiring a lot of inputs. The authors  conclude that Alaska glaciers are in a severe state of disequilibrium, having only undergone 27% of  the retreat necessary to be in equilibrium with today's climate. The observation that the glaciers are  in a severe state of disequilibrium is perhaps not too surprising, and the simplicity of the model and uncertainty of the data used in the model raises some questions about exactly how far the glaciers  are from equilibrium. To me, the more interesting result is the discussion of how the climate  warming scenario affects glacier disequilibrium (the increased warming trend over the past several  decades is much more important than the warming that occurred in the early 20th century) and how  glacier geometry affects disequilibrium. I found myself wondering whether the authors have tried  looking at spatial variations in glacier disequilibrium. I think it could be interesting to see whether  glaciers in certain parts of Alaska are closer/farther from equilibrium.

The main cause of systematic regional variations in disequilibrium would be if there are systematic variations in glacier response times. That will likely outweigh any systematic variations in the time-series of the forced anthropogenic climate response. The response time is generally correlated with glacier area (which we show in Fig. C1, so that would be the best starting point to estimate disequilibrium.

Overall I think this is an interesting and thought provoking paper. I have a few general comments on the paper, mostly aimed at clarifying the glacier model and assumptions associated with it, as well  as handful of specific comments.

**General comments**

1. Section 2 is important for setting the stage for the rest of the paper and laying out the model  assumptions. With that in mind, I think this section needs some additional details and clarifications,  and perhaps some re-structuring.
a) The section immediately starts off with a figure illustrating the idea of committed retreat,  but the figure is based on equations that appear much later in the section. Consider starting  with the model description.

We choose to begin with the figure to visually guide a reader about the concept of disequilibrium and to introduce the definitions we are setting up. We feel that if we dove straight into the model equations before doing that, it would divert a reader's attention away from the core principle. We have extensively rewritten the figure caption, adding a title and much more detail leading a reader through the panels. We've also rearranged much of the introduction text so that the math and figure are closer together. We do want a reader to see a graphical illustration of the disequilibrium before diving into equations.

b) I'm not super familiar with the linearized model, and so in reviewing this paper I spent some time also reading Roe and Baker (2014) and Christian et al. (2018), as well as Roe and O'Neal (2009). It appears to me that the model really originates in Roe and O'Neal, or at least is mostly explicitly spelled out in the appendix of that paper. Consider citing that paper.

Thanks for taking the deep dive! The lineage of our model does begin with Roe and O'Neal, though we think our later expositions were clearer. We now include a reference.

c) I think the model assumptions need to be spelled out more clearly. As I understand it, the linearized model assumes (i) an initial steady-state climate and glacier geometry and (ii) small departures from steady-state. I think the authors are attempting to address (i) in lines 65-69, but not very explicitly and it can be easy to miss this assumption because it is presented a few paragraphs before the model.

We have added a clearer statement that model parameters are taken to be constant over the length scales of interest, and included references to a couple more papers where similar linear models have been used to study climate-glacier linkages.

d) Equation 2 is described as the equilibrium length response to a step change in climate; isn't it just a change in a climate from one steady-state to another?

The reviewer is right - it is the same as going from one steady state to another (although we note there is no such thing as a truly steady state in nature). We describe it as a step-change so as not to have to define the time-dependence of the change in mass-balance forcing. We also use it as an opportunity to make a small point: other disequilibrium studies have assumed an exponential dependence of the glacier response, which gives a significantly different estimate than that of the model used here. By doing it this way, we make the point without calling out specific studies, which is unnecessary. We think it is also a little more mathematically precise to talk about a step change, and then talk about a linear trend.

Then, equations 3 and 4 describe the length anomaly and fractional equilibration for linear climate trends, but the sentence after equation 4 says that it applies for any climate trend. Can these equations all just be written in terms of a general change in climate? It's a bit confusing as is.

We use the linear-climate-trend solutions for some of our analyses, because the physical dependencies on parameters can be more clearly seen. The original language was confusing about which of the parameters gets canceled, thank you. We have also changed the language to be clearer that it is the cancellation that applies for any climate trend, not equation (4) itself.

2. The fractional equilibration (equation 4) depends on the time since the start of the climate trend, the glacier thickness, and the mass balance at the terminus. The authors use $t_0=1880$, ice thickness from regional glacier thickness maps that are known to have large errors, and mass balance profiles from (presumably) representative glaciers. The authors do test the sensitivity of their results to some extent, but I also wonder if it would be useful to use propagation of uncertainty to explicitly show how the model results are impacted by the uncertainty in the model parameters. And I may have missed it, but I don't think the authors tested the impact of their choice of $t_0$ other than to reference another study.

Uncertainty for any single glacier is best judged from figure 9. If a reader has a specific glacier in mind, with perhaps better information than we had for its thickness or terminus mass balance, they can evaluate the value of $f_{eq}$ for their preferred tau. Estimating uncertainty for our distributions of $f_{eq}$ (i.e., across the full population) does not really have an objective approach. If errors among the population are uncorrelated then they would tend to cancel out, leaving the population distribution relatively unaffected. We do report the impact of using different thickness and AAR datasets. We now specifically note that the results are not sensitive to the choice of 1880 as a starting point because, in the realistic scenarios we examine, the bulk of the warming occurs in the last 50 to 100 years.

3. The captions for Figures 3-7 indicate that the panels show the probability density functions and cumulative distribution functions (but note that the caption for Fig. 6 seems to have these flipped). However, I don't think that is the correct terminology. These look like histograms. When you integrate a PDF from $x_1$ to $x_2$, you should get the probability that a sample falls within that range, whereas here you are showing the fraction of the samples that fall within predefined bins. Personally I think the better way to plot this data would be use to empirical cumulative distribution functions and complementary distribution functions. I think histograms can sometimes be misleading when sample sizes are small, such as at the tails of the distributions. Regardless of how you choose to plot the data, make sure that the terminology is correct.

Thank you! We were calculating probability mass, which did not match our labels. We have revised Figures 3-7 so the PDF curves are now calculated to sum to 1, as suggested.

We agree that using empirical cumulative distribution functions (eCDFs) is a valid approach and have tested both methods to confirm the difference is minimal for our dataset (shown below). Because we focus on the population as a whole, we prefer the

binned approach to visualize the distribution's shape at an appropriate level of detail.

[Figure]

Figure R1. A comparison of using binned CDFs (black) vs. eCDFs (red) to display the cumulative distributions shown in Figs. 3, 5, 6, and 7. The top row shows unweighted distributions (blue in the original figures). The bottom row shows area-weighted distributions (orange in the original figures).

**Specific comments**

Title and throughout: This is pedantic, but I think it should be "Alaska glaciers" and not "Alaskan  glaciers" since "Alaskan" is an adjective. For example, see https://www.adfg.alaska.gov/static/home/library/pdfs/writersguide_section6.pdf

We were unaware of the convention! This seems to be a common confusion as we found a mix of usages in other papers. We are happy to contribute to correcting the misusage! We have changed to "Alaska" everywhere.

L12-13 and later: I'm struggling to wrap my head around the area weighting. I understand the equations,  but it's not entirely clear to me what the weighting does.

The area weighting is helpful because of the large disequilibrium of the major bodies of ice that contain most of the glacier mass. This perspective can get lost in the population distributions when the large glaciers get weighted the same as the many very small glaciers. When introducing the area weighting, we now say "Area weighting gives greater emphasis to the larger glaciers in which most of the glacier mass resides, making it more relevant for an assessment of the aggregate mass of Alaska glaciers."

L28: Do I understand correctly that the Johannesson time scale emerges naturally in

your model, which is why it makes sense to use it in this analysis instead of that of Harrison (for example)? Might be good to point that out.

The Johannesson timescale is the standard geometric timescale for alpine glacier response. The Harrison timescale tries to also capture the height-mass-balance feedback, but imposes an extra geometrical constraint and parameter. It is most appropriate for shallow-sloped glaciers. It is worth bearing in mind, but not adopting wholesale.

L42: Models, at least if done correctly, should be spun up with the climate so that they are capturing the current dynamical state.

We agree wholeheartedly! And in our experience, there are variations in the literature in how (and for how long) alpine-glacier models are spun up. It is one goal of ours to highlight the magnitude and variations in the dynamical state of glaciers, which should ideally go into decisions about how to spin up models. We've clarified our intent with this statement.

L46: The response time depends on geometry, so I'm not sure why you are also mentioning geometry here.

This can be seen from eq. (2) or eq. (3). Committed retreat depends on both tau and beta. Beta reflects additional aspects of glacier geometry.

L60: How can they be identical if their response times are different? They must have different geometries and/or climates.

This passage is deleted from the revised text, but there is no problem with having two glaciers that share a common value of beta, but a different value of tau.

L62: "… change in climate from a steady state." ?

We've rephrased this in the revised manuscript.

L68: Suggest "will tend to recover on its own" or "will recover on its own if there is no trend in climate".

Thanks for noting this. We've removed the word 'recover', because it is unnecessary and potentially confusing.

L180: Is it okay to treat \tau as a constant and not something that changes during a warming trend? Maybe something to discuss in Section 2. And also, here you state that you are estimating \tau prior to the start of the warming trend but you are using modern thickness estimates. (I see that you come back to this later in line 271, but again, I think being more upfront about the model and assumptions would be helpful.)

Thank you, the way we did this was confusing. When presenting the model we are now explicit that parameters are assumed constant. We've also added a few more general references of where such models have been used to understand climate/glacier interactions.

We've removed the phrase about estimating tau prior to the warming, which is distracting in that location. The discussion about tau is now in a single place. The bottom line is that uncertainties in H and b_t are more important than the exact moment of time tau is estimated for. We indicated conditions under which the assumption of constant tau is weakest, and we point readers to the section where we consider the impact of uncertain tau on our results.

L280: It took me a little while to remember that f_{eq} was fractional equilibration and to remember what that meant. It could be helpful to remind the reader occasionally; for example, in Figure 7 you could consider indicating that the left side of x-axes indicates more committed retreat and the right side indicates less committed retreat.

Thanks, we agree this would help. We've repeated the definition of f_eq ahead of showing Figure 7, and included how to interpret the limits of low and high f_eq. We also included similar guidance in the caption of Figure 7.

L295: I guess this should be Eq. (4)?

Thank you, good catch!

L323-326: This gets back to my confusion about the model that I discuss in the general comments. Are you using a linear trend for each year? Meaning that Equation 4 is for a linear trend, and so my confusion is related to lines 110-111. So it seems like you are essentially applying a perturbation each year. Are you also then updating the ice thickness and response time? A glacier's response time will change quite a bit as it thins and retreats.

We really, really apologize for this - this was careless equation-referencing on our part, based on legacy drafts. The reviewer is correct that eq. (3) was for a linear trend only, and is right to be confused by what we had. We have rewritten this sentence, and now refer to the correct equation. For a general climate history, we integrate our model (eq. 1) forward in time, using constant coefficients. The impact of those coefficients changing is discussed elsewhere. We've responded elsewhere as to how the revised manuscript clarifies the impact of that assumption on our results. Sorry again.

L431: Do we know how this "dramatic disequilibrium" compares to past climate events?

The degree of glacier disequilibrium is set by the magnitude and rate of climate change. For the typical, global-scale, hockey-stick-shaped climate history over the last two-thousand years, the current disequilibrium would be

unprecedented over that period (see Roe et al., 2021). At the local scale, we lack the information to make such strong statements on millennial timescales.

---

## Author Comment (AC3)

Green = reviewer comments, black = responses

We thank the reviewer sincerely for their time in reviewing our manuscript, and for their extremely thorough assessment and detailed comments and suggestions. This is an exceptionally diligent review, and we appreciate it. We have responded with comments interspersed into their review.

SUMMARY

This study investigates the state of disequilibrium of ~5600 glaciers in Alaska using a well published "model"/equation that links mass-balance anomalies and retreat in terms of length. The  analyses focus on how different response times and the shape of the climate forcing affect the  state of disequilibrium. Overall, they find the glaciers are in a state of severe disequilibrium and  thus a lot of retreat is already inevitable in the future.

Overall, the study is highly relevant and provides new knowledge about the state of Alaskan  glaciers, thereby making it suitable for The Cryosphere. However, there are considerable areas of  the manuscript that could be revised to improve readability and more clearly convey these  findings. Specifically, the method descriptions were fairly hard to follow, even though the  concept itself is fairly "straightforward" (i.e., there's a model that can determine the state of  disequilibrium and this is stated with the f_eq metric). In fact, the "straightforwardness" of the  analysis is one of the most eloquent pieces of the analysis; hence, if the methods and readability  can be improved, I think it will reach a much larger audience and the key points will come across  much better. I thus recommend major revisions.

**MAJOR COMMENTS**

Section 2 is a bit hard to read, which seems to be due to the structure of the section. For  example, the first sentence refers to Figure 1, which ultimately took me to read Figure 1;  however, without any context, it was very hard to understand. Another issue with this section is  that Figure 1 is technically a result; however, there's no information about the model and  methods given yet – that all comes later. It thus feels very confusing and forced me to read it  twice in order to understand, i.e., understand the methods first, and then understand this  theoretical result. It also seems to provide remarkably detailed justification related to not  focusing on natural interannual climate variability without any context. If you want to keep this  theoretical case (i.e., results) as a way of describing the concept within this section, I would at a  minimum recommend that you restructure the section so that the text is logical to follow in a  single read. I would further recommend that you focus on the key elements, and provide detailed justification as relevant later on.

Thank you for this. Both reviewers brought this up, and we agree we could have been

less confusing for readers. We've restructured and rewritten this section. We do want to start off with the description/illustration of disequilibrium, rather than beginning with the model, in order to begin with a focus on the concept. The revised version:

- Goes straight to Fig. 1, and clearly walks a reader through what disequilibrium is, and introduces terms that will be used later.
- Rewrites the caption of Fig. 1, and adds a title to be much clearer about what we want a reader to get; and leads the reader through each of the panels.
- Removes extraneous material about natural variability etc. to later in the text.
- Introduces the model specifically as a tool for calculating these disequilibrium factors.

We think this improves the flow and transitions considerably. While we do use the model equations to create the curves in Fig. 1, the concepts are general and not limited to the model.

Section 2 and 3 – check on formatting requirements, but it's a bit odd to see equations that are written in both equations as well as equations written in lines (e.g., L96, 97, 105, L158, etc.).

We see past articles in this journal that include both in-line and separate enumerated equations. In our experience it is common to put minor equations and those not referenced later as in-line equations.

The heavy use of variables in place of actual text reduces readability. Even something as simple as writing out tau as "glacier's response time" or just "response time" or $H$ as characteristic thickness could greatly improve accessibility and readability of the manuscript and only add one word. This is also likely part of the reason Section 2 is challenging to read.

Hopefully the layout of Section 2 is now clearer, and that symbols are now introduced without other distractions. It is our standard practice (which we hope we have adhered to consistently) to use symbols where a variable or other mathematically precise term is used repeatedly in the main body of the paper, and to use them consistently thereafter. There are dozens of places where tau and H are used. As we think is standard, we avoid using symbols in the abstract and introduction; and then make sure they are redefined in the summary and discussion. We've also aimed to use symbols that are cognitively linked to what they describe (H,tau, etc.). If the reviewer or editor feel strongly enough, we could introduce a glossary of terms used. It is our feeling that we are not over-the-top in our use of symbols, relative to other similar work.

Can Figures 3, 5, 6, 7 be merged together? This would be nice to able to see the differences. Also, it's interesting that the regional data is provided in Figure 2; however, these regional differences aren't really included within the analysis. It could be interesting to see Figures 3, 5, 6, 7 with these regional colors included within the pdfs. I recognize this would not enable the area-weighted and count pdfs to be done, so there's some discretion as to the important narrative to show. Maybe there could

be a Figure 3, 5, 6, 7 pdf with just the counts in the  supplement/appendix?

We understand the point here. However, because the figures follow the development of the analysis, we feel it would be more confusing for the reader to encounter information that has not been introduced yet. We've aimed to help readers make comparisons by providing all the figures in a common format.

We've chosen to do Alaska as a whole in the main paper. It would be an extra layer of analysis to show the results by region, and would not necessarily be more insightful. We would have to have separate panels for the PDFs and CDFs, etc.; and we might comment only on a small subset of the extra curves. A reader can get a broad sense of the regional breakdown already because we present a table of results broken down by glacier area (Table A1), and the distribution of areas among the various regions is presented in Figure 2.  We've added a sentence in the Summary and Discussion pointing this out. So thank you for the suggestion.

There are many parts where it seems like results are provided in the methods and methods are  provided in the results. See specific comments below, but these really break up the flow of the  text.

We've chosen not to follow the I-M-R-D-C recipe for paper structure. It is a style preference for us to put methods close to the results that use them. Our apologies if this is jarring! We agree with (both) reviewers that Section 2 was structured in a confusing way, and hope the revisions are an improvement.

Discussion: I was surprised not to see any comparison to prior values of the state of disequilibrium or committed mass losses? It seems like referencing these studies, especially  those in Alaska (e.g., Davies et al. 2024, *Nature Communications*) is needed at a minimum. I  suppose that this model can only do length changes and not mass change. This may be an  important distinction that should be highlighted in the discussion?

The Davies paper has a different aim from ours, we think. It has a specific process focus, and we don't see how it addresses committed retreat (or committed mass loss) in a comparable way; and a reader has to infer that there is disequilibrium rather than it being directly addressed. This comment prompted us to include a citation Mernild et al. (2013), and to Zekollari et al., (2020) which explicitly estimates committed mass loss for the Alps.

There is a general understanding that glaciers must be in disequilibrium, and obviously there are lots of studies that characterize observed rates of retreat, and project continued retreat in the future. But it is an extra step to estimate disequilibrium specifically.

The glacier-wide climatic mass balance is the response of the glacier to the climate. The  equilibrium state is thus the geometry that is needed to support a given

climatic mass balance  gradient.

We are not quite sure that we understand the point the reviewer is making. The glacier-wide mass balance is partly a function of the glacier response, and partly a function of a changing climate. It definitely is a good complementary way to think about the state of (dis)equilibrium, and we'd say the equilibrium geometry is that which gives a glacier-wide balance of zero (assuming that's what is meant by "support"). We focus here on length disequilibrium, which lends itself to analysis when only large-scale geometry is available in the inventories.

LINE COMMENTS

L32 – consider replacing the "-" with "as" to make it easier to read.

We've tweaked this in the revised manuscript.

L42 – This sentence feels out of place as it's a result but stated as a "for example" in the  introduction. I don't think it's needed and distracts more than helps.

 We appreciate the reaction, but choose to keep it in. We follow the guide style that an introduction should include mention of the results. And we wanted to give a reader some quantitative guidance as to what we mean by "severe disequilibrium".

L55-56 – consider rephrasing. The "…, and for three different assumed …" doesn't make sense.

We've made this change in the revised manuscript.

L70 – this sentence describing the equilibrium length, if I understand this correctly, then this is  the length if you were to hold the temperature anomaly at any given point and allow the model to  run until it reaches it's equilibrium length? If so, some mention of the timescales being different  (i.e., it's a seemingly infinite timescale) is important to note here; otherwise, it looks like the  glacier retreats that length immediately which does not make sense.

The reviewer is correct that L'_eq is the equilibrium length change in response to the (fixed) climate change. We are not sure what the reviewer meant by different timescales here. The glacier asymptotes to its new equilibrium at a rate governed by its response time, as can be seen in Fig. 1 when the warming ceases. Hopefully this is all clearer with the restructured section 2.

L70/71 – length anomaly should specify what the anomaly is calculated with respect to, i.e., the  initial length.

We replaced the word 'anomaly' with 'change', as it is easier to understand at this point of the manuscript, and is unambiguous.

L71 – consider replacing the ":" with "as" to make it easier to read.

Thanks, we switched to 'because'.

L74 – why use years at all? The use of years implies it's associated with a specific time period. If it's intentional, i.e., that it reflects the change in temperature from pre-industrial, then this should be stated along with the justification. At the same time, I suspect that this would become exceptionally tricky here given Arctic amplification; hence, you have variable increasing temperatures for different areas, so the justification should make it very clear (and explicit) as to how it's being selected.

Hopefully this will be clearer in the revised section because there is now less extraneous information. We now state we are considering an "idealized industrial-era warming scenario" starting in 1880, so we think it should be clear that our scenario is inspired by the realistic situation. The detailed amplitudes of warming are not important for the principle of disequilibrium, but give a reader a sense of the kinds of magnitudes being discussed. With the revisions we don't think a reader is going to be disconcerted by the mention of specific years.

L78 – I don't know if this level of detail is necessary as the concept is fairly straightforward, but for my own curiosity, what happens for overshoot scenarios? I assume the current text is assuming only scenarios of consistently increasing temperatures?

Temperature trends in this work are monotonically increasing, with the exception of the period 1940-1965 in the ETCW scenario. We've considered discontinuous-warming scenarios using the same model in other work (e.g., Fig. 4 of Christian et al., 2022; Roe et al., 2021). The overshoot scenario is interesting to consider and we expect response times and current disequilibrium would affect the response in such a case, but our focus here is on the dynamic state with respect to past warming, rather than potential future scenarios.

L80 – regarding "increase their rate of change until it matches that of the warming trend", how is that possible given they're completely different units (e.g., m/yr versus degC/yr)?

Thank you for pointing this out, the phrasing indeed implies an impossible comparison! The revised text is now dimensionally consistent.

L80 – "and so asymptote to as" doesn't make sense. This reads as though asymptote is a verb. If I understand what's being implied it's referring only to the orange line? The current sentence reads quite general so I was very confused since the 75-year

response time example certainly has  not reached its new equilibrium in this example.

We would maintain that asymptote can be  used as a verb in this mathematical context, but for clarity have changed to "approach". We've changed the figure citation to refer specifically to the orange line.

L80 – this also assumes that the glacier is large enough, i.e., there's enough mass loss (or length  change remaining) to enable this.

The reviewer is right, of course. We think that readers will take this as a conceptual illustration of disequilibrium. We mention the impact on glaciers with dramatically changing geometry, and exclude the smallest glaciers from our study because of these kinds of concerns.

L81 – choose a different word than "spun up" as this implies initialization conditions of a model,  which is not the case here.

Changed to "fully adjusts" to remove the ambiguity.

L90 – "circumventing uncertainty in some inputs" is clearly model specific. I would state this as  other readers may be interested in applying these metrics to other models (for example) and thus  this might not apply. Note: see major comment as we still don't have any knowledge of "some  inputs" yet, so this is also confusing to read.

Thanks for this. There is a general tendency for the impact of some uncertainties to be minimized, but the cancellation is only exact for a linear model. We've removed "input" and clarified the language.

L102 – can you add the context as to which metric (length, time scale, mass) this "estimate"  refers to?

It is disequilibrium in glacier length, now stated.

L106 – "… within 5% of the asymptotic limit (e.g., Fig 1c, orange line)" – this is wonderfully  clear! This type of statement following the description of the method is very valuable and I'd  encourage it to be earlier.

Thank you. Hopefully the rest of the Section reads more clearly now. We get to the model much faster in the revised version.

L107-108 – This reads as an important result, but in a conceptual/methods section. Suggest  moving to the results section.

Thanks for this. We included it here because it is the first time we are talking about the phases of adjustment, and it gives readers something to connect with later. We now also repeat this point in the summary and discussion, as an implication of our

results. We agree it is worth emphasizing.

L109 – is this different than the "climate trend" described by L105? If different, specify this. Equation 4 – why is there a box around this?

Thanks. It is the same trend. Language tweaked for clarity. Box removed.

Figure 1 – can you show the initial length on one of these figures? It'd be nice to see the raw  metric as opposed to only the anomalies/metrics.

The initial lengths are actually not defined for these calculations. We set the value of beta = 100 (included in the caption). beta  = area/(width x thickness) of the initial state, so it applies to any combination of those that  gives 100. For a uniform width of 1 km, and a thickness of 200 m, it would be a 20 km long glacier.

L118 – "retreat" not "retreats", no?

We prefer 'retreats' (plural) deliberately to emphasize that each glacier exhibits its own distinct retreat behavior with unique characteristics. Either would make sense in this context but 'retreats' confers a slightly different meaning.

L119-120 – again, it would be useful to show the metrics (and potentially state the differences)  instead of only stating the differences here.

We're not quite sure what the reviewer means by showing the metrics. The metrics are shown in the figure. But in terms of magnitude, the fractional equilibration does not depend on beta. The magnitude of 3 km does depend on that choice for beta, but as noted above that value can apply to a family of different glacier geometries. So the intent is for those numbers just to be illustrative.

L137 – What about debris-covered glaciers? I assume their retreat behavior would complicate  this?

It is a fair point. Maybe it would be possible to screen for degree of debris cover, but debris cover exists on a continuum, so it is hard to know how to filter it objectively. Debris cover would most likely impact tau via its influence on $b_t$, but in ways that are hard to account for. We now mention this briefly in Section 3 as a potential filtering criterion, so a reader is at least alerted to this issue.

L139 – This would read better deleting or changing "Lastly" since it refers to the last two.

Thank you, we agree. We've changed this in the revised manuscript.

Figure 2 caption – I assume the color-coding in Figure 1b and 1c refers to the subregions. It'd be  good to explicitly state this.

The color schemes in Figs. 1 and 2 are unconnected. We don't think there is a substantial risk of misinterpretation. The captions of each figure are clear about what each color represents. Note that the caption for Fig. 1 has been substantially revised, reducing the chance for misinterpretation.

Figure 2 – Why not use something different besides circles for the long-term mass balance  records to avoid confusion?

Thanks, we have changed the shape in the revised manuscript.

Figure 2 – I assume you're aggregating these areas based on the center lat/lon? It looks like a  regular grid of some sort. Would be good to state the resolution of this grid.

We use a hexagonal grid (displayed as circular markers for visual clarity) with each marker positioned at the center coordinates of its corresponding grid cell. We have added a concise description of this as a note in the caption.

L154-157 – perhaps good reason to add a CDF (even if it's just the total) to these two plots as  well as a right-hand side axis.

We appreciate the suggestion. However, taken together, along with the CDFs included in all the other results, there is a risk of information overload. These two lines of text basically serve to motivate why we present all our results as both number-weighted and area-weighted, PDFs and CDFs. The two included panels together provide intuition about the weighting of the population towards large-area glaciers. We think we'd have to add two new panels with regionally distinguished contributions to the CDFs of the numbers and areas to be consistent. We aren't clear that that would be worth it.

L157 – delete "simple". It's a glacier count. This implies glacier area is complex somehow?

We've revised the text to be more explicit about number weighted and area weighted.

Section 4.1 – Shouldn't this be "Characteristic thickness"?

Thank you! We have made this change.

Figure 3 – change "the method outlined in the main text". If you want to highlight this, state (see  Section X) or something similar.

Thank you. We now refer to the specific section.

L209 – I assume you're referring to Order 2 RGI regions? If so, state this.

We mean the glacierized regions indicated in Fig. 2 (text tweaked in the revisions), indicated by the points in Fig. 2.

Figure 4 – Is there a reason that you include the accumulation areas (i.e., > 0 m/yr) when the methods state you're only using the ablation area data? Does it change the numbers you get at all?

The figure documents the available data, and does provide a sense of how constant vertical gradients in mass-balance are, across the ELA zone. Since we are picking a region-wide single number, the figure shows that number is not sensitive to those details.

L233 – no need to repeat the caption. Recommend re-writing such that you just state what the b_t values are and then refer to Figure 5a,b in parentheses.

In most cases, we prefer a direct citation style for figures, although we recognize that preferences differ. In our view, we prefer to lead a reader through a figure, and directly indicate points of focus. The indirect (parenthetical) style expects readers to make some steps by themselves, which risks ambiguity.

L250 – Figure 6 shows … is just repeating the caption.

As noted above, this is our preferred, direct citation, style.

L251 and others – I don't think you need to state (and 90% range) every time. Only at first use.

We prefer to retain this style. We recognize that it is repetitious, but want to direct a reader's attention to this range, as it carries important information about the width of the distributions.

L264 – can you give the areas or just a broad definition here of how "small" is used? < 10 km2 for example?

Thank you for this suggestion. The reviewer's suggested threshold of 10 km² aligns well with the glaciers we describe as 'small' in this section. We've added a note in the text to specify this.

L299-314 – this is all methods.

As noted above we are not following the I-M-R-D-C structure. We prefer to introduce methods close to the analyses and results that use them.

Figure 8 – suggest using different colors to avoid confusion with the same colors being area weighted vs. count in Figures 3, 5, etc.

Thank you for this suggestion regarding Figure 8. We agree with the reviewer's

suggestion to use different colors in Figure 8 to avoid confusion with the color schemes used in Figures 3, 5, etc. We will revise Figure 8 (and Figure 9) with a distinct color palette to clearly differentiate between the three warming scenarios and the area-weighted vs. count-weighted distinction used elsewhere.

L319 – remove "see text for details" as that's obvious. Refer to a section if essential.

We agree and will remove the redundant phrase from the figure caption.

L329 – Recommend not using highly unique acronyms like "ETCW" and "GWI" as they greatly impede readability. If writing three to four words is deemed too large, then come up with a new name to refer to them as that's meaningful.

We choose to retain the initialisms. They are closely connected to what they describe, and they are standards used elsewhere in the literature. Different names would be confusing in that regard. There are 10 usages of ETCW and 15 of GWI. That is frequent enough to merit defining initialisms, we think. Our preferred style is to introduce initialisms if they are used frequently enough, and then to be completely consistent with their usage thereafter.

L336 – The shape of the warming is important compared to the linear shape; however, the "ETCW" and "GWI" shapes provide relatively consistent values. This seems worth mentioning earlier (noted it's in L341-342) since it currently reads that they're all highly different until you get there.

Thanks for this reaction, although we don't think we share it. The two ETCW and GWI scenarios are common in the literature, and merits exploring them both. We guide a reader to Figure 8 starting at 327 (in the submitted manuscript), right after describing how we calculate $f_{eq}$. We immediately begin to describe the commonalities in $f_{eq}$ of the GWI and ETCW scenarios, so we are not sure where we create the impression that the results are all highly different. The similarity of the GWI/ETCW results is a main theme for the remainder of the paper, and is noted (albeit in an indirect way) in the abstract.

L336 – Building on the impact of the shape, I would recommend analyzing this relative to the linear $f_{eq}$, i.e., reporting the differences on a per glacier basis. This would make it a lot easier to grasp the statements like L338-340. L330-334 report the 90% range, which is highlighting the glacier variability, but ultimately this whole paragraph is about the differences on a per-glacier scale.

We think the reviewer is suggesting we take the differences between the linear $f_{eq}$ and the GWI/ETCW $f_{eq}$ for each glacier and plot the PDF distribution of those differences? If that is right, a plot of differences is going to scramble the data in ways that are hard to interpret. For instance, from Fig. 8 it can be seen that differences are small at both low $f_{eq}$ and high $f_{eq}$ (long and short response times, respectively). But if the distribution of differences is plotted, that distinction will be lost. We already show

and describe how the warming scenario affects f_eq in Figure 9. So a reader can take the tau of their favorite glacier and see the impact of warming scenarios on disequilibrium.

Thank you! This comment caused us to reexamine what we had written, which we then realized was actually wrong. The biggest differences in f_eq occur at times when one scenario has experienced cooling (f_eq calculated after a forced cooling can exceed 1 - see attached figure). We deleted the original text, and added a short sentence with the new information.

[Figure]

Figure R1. Comparison of f_eq over time between the ETCW (solid lines) and GWI (dashed lines) warming scenarios across an illustrative range of tau (colors). f_eq is identical between warming scenarios up until 1940 because both trends are linear and monotonic over the initial period (see Eq. (4) in the submitted manuscript).

Thank you! We've corrected the issue in the revised version.

Figure 9 – This looks highly smooth. I assume some function was fit through the data? Please state this in the caption.

These curves are solutions to the model equations, so no data was used. The solutions are smoothly varying functions of tau (see, e.g., eq. (4), which provides the solution for the linear scenario. We added a note in the caption pointing the reader to where we described the method.

L356 – what information is not available? This is quite vague.

We mean only that objective uncertainties do not exist for the thickness or terminus balance rates. Assumptions of what is reasonable have to be made. So we chose to instead simply apply deliberately large uncertainty bounds for both parameters to demonstrate that even under conservative assumptions, our core findings remain robust. This approach allows us to show that the conclusions are not sensitive to the availability of these specific uncertainties.

L356-359 – is this not methods?

As noted above, our style is to intersperse methods throughout the paper.

L350-367 – prior to this section, there are a lot of values provided. Then in this section, there are no values to actually accompany the uncertainty estimates. Instead the uncertainty is just qualitatively described and not shown. Can one provide some values? Maybe for some of these different examples?

We apologize if we are not understanding the point. The uncertainty is quantitatively shown in Figure 9. We are asking a reader to look at Figure 9 to see how uncertainty in tau (i.e., variations on the x axis) and uncertainty in warming scenario (variations among the three lines) affect our target f_eq (shown on the y axis). We provide a reader with our opinion of what a reasonable uncertainty on tau is; we point out specific taus at which uncertainties maximize; and discuss the impact in the small- and large-tau limits. We've added specific guidance in the caption as to how to use the figure to estimate uncertainty in f_eq.

369-375 – Like my prior comment, the interesting thing here is the difference by glacier, not the actual values. Consider changing the way this is provided. For example, are all glaciers 0.08 higher or is it highly variable? Is there not a figure that shows this that can be referenced as well?

Recall this is a population of 5200 glaciers. So a different dataset for H yields some thicker, some thinner. We do describe the general differences in the population distributions of H, and AAR from the different datasets, and then we propagate that into how it impacts the distribution of f_eq for our population. We don't know how to present this glacier-by-glacier. We've chosen to assess the whole population in this study. In the discussion we defend this approach and note that, for any single glacier, we would recommend a more thorough assessment of local information.

L380 – suggest pulling out a few important numbers from the table here to make this more quantitative. Again, it'd be interesting to know the differences, not the values for all of them (see prior comments).

We apologize for being at cross purposes here, but our reaction is the same as above. We think showing the distributions of H, bt, tau in earlier figures helps to illustrate

heterogeneity between individual glaciers. But our chosen goal is to assess the population, and we report sensitivity to different datasets in the population statistics. The essential result is that glaciers are in a severe state of disequilibrium, and it is robust to these uncertainties.

L389-392 – Are these statements supported anywhere? Figure 9 does not seem to show this.

Thank you for this comment. We moved most of this material in this paragraph to the appendix. We agree that these details are not needed in the main body of the paper.

L406-411 – Consider rephrasing. It's a bit odd to state that more detailed numerical modeling  can address a problem but then state that you don't expect it to be any different. In that case,  there's not really a point in having a more detailed numerical model.

We say it might be applied, but we are also aiming to add a note of caution. There is currently a tendency to use ever-more complicated numerical models, but in the face of uncertain inputs it does not guarantee a better answer.

L445 – "other uncertainties in the setting" – provide some examples. Otherwise, highly vague.

Thanks. We've been more specific.

L458 – "It seems important" consider making this line stronger. Awareness of how much retreat  is already baked into the future evolution of glaciers is important for policy makers, resource  managers, and the general public to know to _____. Perhaps that _____ is preserve the glaciers  that remain? Reduce our greenhouse gas emissions?

We reflected on this. But we think a stronger statement would be verging on advocacy, which we don't think is appropriate in a scientific article of this kind. What the public or policy makers choose to do with an understanding of this disequilibrium should be up to them. We see our role is to report on the information.

---

## Referee Report (RR1)

The authors have done a very nice job of addressing my previous revision. I still have a few very minor comments, but otherwise I feel that this paper is ready for publication.

L67: Figure 1a refers to a temperature change as \$\Delta t\$ and not \$T^\prime\$. I think they are referring to the same thing?

L75: Does the absolute value sign have some meaning? Isn't  $L^\perp = L^\perp =$

L78: "matches" or "approximately matches"? Is this referring to line 110?

L87 and elsewhere: I am still unsure about the use of the terms PDF and CDF for the plots that you've generated. (I'm not an expert in probability and statistics, so I could be wrong about this. If so, my apologies!) My understanding is that when you integrate a PDF from x1 to x2, you get the probably that a randomly selected sample will fall between those two points. Let's say you have a bin that goes from 0 to 100 and has a value of 0.001. If you integrated from 0 to 50, you would find that the probability of a random sample falling between those two numbers is 50\*0.001=0.05. Similarly, if you integrated from 50 to 100, you would also find that the probably of a random sample falling between those two numbers is 0.05. So essentially you are saying that the probability doesn't change within a bin, but then undergoes a step change at the edge of the bin, which seems strange. This is why I suggested using empirical CDFs and CCDFs, which don't have that issue. Maybe the graphs should be referred to as histograms or binned PDFs? Regardless, any changes you make here wouldn't affect the outcome of the paper.

L104: Is this necessarily a step-change, or just a change from one steady-state climate to another?

L151: Is it possible to say in a half-sentence or so why your model doesn't work for small glaciers? Is this essentially referencing Fig. C1?

L172-174: In my initial review I had a comment about not fully understanding the area weighting. I think it makes sense now. When discussing response times, for example, I think the area weighting essentially tells you the response time for a randomly selected area of glacier ice. Since you are more likely to select a random area from a large glacier, that will shift the PDF and CDF to the right. Perhaps it would be worth adding a sentence to this effect?

L289: Isn't \dot{b}\_t the only other thing affecting \tau?

L384: Should this be "We consider this to be a reasonable estimate"?

Fig. 9: Have you tried including the uncertainty in \tau in the plot? I wonder if it would be easier to understand the uncertainty if there were shaded regions.

L387: "we assess"

---

## Referee Report (RR2)

**SUMMARY**

This is my second time reviewing the manuscript. This study investigates the state of disequlibrium of ~5600 glaciers in Alaska using a well-published "model"/equation that links mass-balance anomalies and retreat in terms of length. The analyses focus on how different response times and the shape of the climate forcing affect the state of disequilibrium. Overall, they find the glaciers are in a state of severe disequilibrium and thus a lot of retreat is already inevitable in the future. The manuscript is suitable for the Cryosphere and contains new knowledge about Alaskan glaciers.

The revised version of the manuscript is much easier to follow, albeit at times a bit redundant between description in figures and text. That said, in the first round, I had to read the article twice to understand what was done. This second round of review, I have the benefit of already reading the article twice, so it's hard to know how much improvement is from the slight changes they made to the sections compared to being familiar with it already. I'll also note that the response to many comments in the last round was that things were related to the authors' preferred writing style (e.g., use of acronyms, variables, in-line equations; not using a traditional convention of sections but rather merging results in the introduction, methods, etc.). I'll defer to the editor on whether these changes should be made or not. In my opinion, they greatly hinder readability, and I've highlighted some examples of these below so the editor can quickly assess for themselves.

Assuming the writing style/format of the manuscript is suitable to the editor, I thus recommend minor revisions.

**Major Comment**

L433-435: It should not be on the reader to determine regional variations. Furthermore, as the authors noted in their response to other comments, they have chosen to deliberately "guide a reader" through various aspects of the study but then leave this entirely to the reader. Specifically, they are now asking a reader to (i) look at a table in the appendix that breaks down various metrics as a function of size, (ii) go to Figure 2 and use the color-coding in Figure 2a paired with Figure 2b to get a sense of the regional variations in size, and (iii) to form an interpretation of what these regional variations look like. I have spent a couple minutes trying to do this and it's borderline impossible to infer anything meaningful. If regional variations are an important aspect of this study, then there should be a short summary here of the main takeaways. Even better, is to provide a table in the appendix that explicitly breaks things down by region. If the authors do not deem this to be important, then remove this reference or state that future work could investigate regional variations or something similar.

**Specific Comments**

L53-56: The authors have chosen to keep results in the introduction. I don't understand the rationale for doing this, so defer to the editor on whether this is appropriate or not.

L56-61: The stark contrast between "traditional" writing practices (e.g., ending the introduction with a statement on what each section does) and "non-traditional" writing (e.g., mixing results with the introduction, methods, etc.) is a bit jarring. This was noted in the last round of reviews, so I defer to the editor on whether this is appropriate or not.

L75: are absolute brackets needed? It seems preferable not to be absolute in order to show the direction of change.

L84: remove the comma in the citation.

L97: delete the comma after the colon.

L163 or L167: If these are Order 2 RGI regions, state this explicitly.

Figure 3: is there a reason not to use "cumulative dens." To be consistent with the PDF in (a)?

L299-306: here are good examples of where in-line equations look odd and reduce readability. Rather than write an equation for L300-301 that  $f_eq = 0.57$ , one could write "The median fractional equilibration is 0.57 meaning the average glacier still has ..."

L301: "still has still has"

L320-336: a good example of (i) the text repeating the figure caption and (ii) a considerable amount of methods in a "results" section. I understand the authors' desire to put relevant methods near results and not use a traditional layout. In my opinion, this writing style becomes redundant and breaks up the narrative. Again, I simply mention here to point out to the editor as the authors have already made their opinion clear.

L365: appear to be missing a space after scenario,

Figure 9: move to Section 5.3 when it's first used. I assume this will be done in a typset.

L384 & Figure 9: given the explicit mention of uncertainty, can you add the uncertainty onto the figure?

---

## Author Response (AR2)

**Prof. Marzeion,**

We have completed revisions and responses to the second round of review. We appreciate the reviewers' willingness to do a second round, as well as your input. In addition to addressing specific and technical comments, we worked through the manuscript with an eye towards improving overall readability. We regret if our initial responses and revisions didn't adequately address these concerns – we agree there were still some areas to improve clarity. We think our overall structuring (which admittedly differs from an explicit methods-results-discussion format) can work well for integrating a theoretical framework with data analysis as done here – that being said, we acknowledge this choice requires special attention to readability to avoid losing the reader. So we appreciate the nudge to revisit these issues.

Our changes are detailed in the response document, but in brief, we have clarified our structure "roadmap" closing the introduction; revisited figure captions to minimize redundancy with text, and have reviewed instances of inline equations and expressions for clarity, fleshing out some expressions with text where necessary.

We have sought to address the spirit of the concerns raised by reviewer #2, which we hope have improved overall readability. On some matters we've retained our preferred style, such as the use of inline equations, but have reviewed these for clarity. On these matters we believe we are within the bounds of previous publications in TC, but of course defer if we've missed any specific house rules here.

Finally, we have responded to the more technical points raised by both reviewers, which we also think improve clarity. In particular, we have added some annotation to Fig. 9 which we think helps with visualizing uncertainty. Reviewer 2 also highlighted an issue related to (sub)-regional variations. We recognize that readers will be interested in this information, so have added Table A2 to the appendix, which presents our key results categorized by RGI second-order region, along with a brief description of regional variations in Appendix A.

We hope these revisions address the remaining issues and we look forward to your assessment. Thank you for your consideration.

On behalf of the authors,

**Responses to Reviewer 1**

Green = reviewer comments, black = responses

We thank the reviewer for their constructive second review and for confirming that our revisions have adequately addressed their concerns. We appreciate their attention to detail in identifying the remaining minor issues, and we have addressed each of their specific comments. We are grateful for their contribution to improving the clarity and accuracy of the manuscript! We have responded with comments interspersed into their review.

The authors have done a very nice job of addressing my previous revision. I still have a few very minor comments, but otherwise I feel that this paper is ready for publication.

We are grateful to the reviewer for their time and contributions in providing a second review.

L67: Figure 1a refers to a temperature change as \$\Delta t\$ and not \$T^\prime\$. I think they are referring to the same thing?

Thanks. The reviewer is correct that these both refer to temperature change, but our notation in the figure caption was inconsistent with the text. We've corrected the caption to use T'.

L75: Does the absolute value sign have some meaning? Isn't L^\prime – L^\prime\_{eq} greater than or equal to 0 when you are talking about committed retreat? Or is that |L^\prime – L^\prime\_{eq}| is the disequilibrium, which just happens to be positive when a glacier is retreating?

We've included a sentence explaining our choice to readers. Retreats are negative numbers, and subtractions of negative numbers can cause temporary confusion as a reader has to go through the step of subtracting a bigger negative number to get a positive number. The use of absolute brackets saves the reader this step. Moreover, it might also confuse the reader that the "committed retreat" is a positive number, since retreats are negative. We now concretely state that it is the magnitude of the difference we are interested in. The use of brackets at all makes it a grouped term, which is useful since we refer to it often in the paper.

L78: "matches" or "approximately matches"? Is this referring to line 110?

This indeed refers to line 110. Put another way, dL/dt - dL\_eq/dt approaches 0 as t -> inf. We have reworded slightly to clarify that this exact match of retreat rates occurs in

the context of the linear model, though it is generally the case that retreat initially lags but then accelerates.

L87 and elsewhere: I am still unsure about the use of the terms PDF and CDF for the plots that you've generated. (I'm not an expert in probability and statistics, so I could be wrong about this. If so, my apologies!) My understanding is that when you integrate a PDF from x1 to x2, you get the probably that a randomly selected sample will fall between those two points. Let's say you have a bin that goes from 0 to 100 and has a value of 0.001. If you integrated from 0 to 50, you would find that the probability of a random sample falling between those two numbers is 50\*0.001=0.05. Similarly, if you integrated from 50 to 100, you would also find that the probably of a random sample falling between those two numbers is 0.05. So essentially you are saying that the probability doesn't change within a bin, but then undergoes a step change at the edge of the bin, which seems strange. This is why I suggested using empirical CDFs and CCDFs, which don't have that issue. Maybe the graphs should be referred to as histograms or binned PDFs? Regardless, any changes you make here wouldn't affect the outcome of the paper.

We appreciate the close attention to this! This understanding matches ours. We chose to keep the figures as histograms because we feel they make it easier for a reader to do this probability calculation by eye, and to see the distribution of density throughout the range. Per the reviewer's suggestion, we added a note at the introduction of these plots to clarify that they are histograms.

L104: Is this necessarily a step-change, or just a change from one steady-state climate to another?

This is a good distinction. The solution for L\_eq indeed refers to any transition between steady-state climates. However, for the transient response, the sigmoidal shape mentioned is specifically for a step change. We've adjusted the wording to remove this stipulation where unnecessary.

L151: Is it possible to say in a half-sentence or so why your model doesn't work for small glaciers? Is this essentially referencing Fig. C1?

Yes, and we've added it! This actually was not meant in reference to Fig. C1, but the reviewer points out an interesting connection.

L172-174: In my initial review I had a comment about not fully understanding the area weighting. I think it makes sense now. When discussing response times, for example, I think the area weighting essentially tells you the response time for a randomly selected area of glacier ice. Since you are more likely to select a random area from a large glacier, that will shift the PDF and CDF to the right. Perhaps it would be worth adding a sentence to this effect?

The reviewer has the right idea, but we would frame this differently, as it is difficult to conceptualize the response time of a small component of a glacier (it better describes the full system response). If you were characterizing the effective response time for a population of glaciers, however, you might want to take account of the fact that a larger fraction of the population's area comes in bigger glaciers with larger response times. The area weighting reflects that.

L289: Isn't \dot{b}\_t the only other thing affecting \tau?

Our estimates of both H and b\_t are affected by using the modern glacier geometry. However, this error is minor compared to the larger uncertainties in estimating glacier thickness and terminus mass balance, particularly for large glaciers.

L384: Should this be "We consider this to be a reasonable estimate"?

Yes, thank you.

Fig. 9: Have you tried including the uncertainty in \tau in the plot? I wonder if it would be easier to understand the uncertainty if there were shaded regions.

Both reviewers made this suggestion. We hope we've found a way of conveying that on the new graph in a way that is helpful and not distracting.

L387: "we assess"

Nice catch! Thanks.

-

**Responses to Reviewer 2**

Green = reviewer comments, black = responses

We thank the reviewer for their thorough second review and for their patience in working through this manuscript again, and we have taken their additional feedback seriously. In this round of revisions, we have worked to address their concerns about readability while maintaining our analytical approach. Specifically, we have added Table A2 with regional breakdowns as requested, reduced redundancy between text and captions, and improved the flow of inline equations and expressions. We have responded with comments interspersed into their review.

**Summary:**

This is my second time reviewing the manuscript. This study investigates the state of disequlibrium of ~5600 glaciers in Alaska using a well-published "model"/equation that links mass-balance anomalies and retreat in terms of length. The analyses focus on how different response times and the shape of the climate forcing affect the state of disequilibrium. Overall, they find the glaciers are in a state of severe disequilibrium and thus a lot of retreat is already inevitable in the future. The manuscript is suitable for the Cryosphere and contains new knowledge about Alaskan glaciers. The revised version of the manuscript is much easier to follow, albeit at times a bit redundant between description in figures and text. That said, in the first round, I had to read the article twice to understand what was done. This second round of review, I have the benefit of already reading the article twice, so it's hard to know how much improvement is from the slight changes they made to the sections compared to being familiar with it already. I'll also note that the response to many comments in the last round was that things were related to the authors' preferred writing style (e.g., use of acronyms, variables, in-line equations; not using a traditional convention of sections but rather merging results in the introduction, methods, etc.). I'll defer to the editor on whether these changes should be made or not. In my opinion, they greatly hinder readability, and I've highlighted some examples of these below so the editor can quickly assess for themselves. Assuming the writing style/format of the manuscript is suitable to the editor, I thus recommend minor revisions.

We thank the reviewer for their time in doing a second review. We are glad that the revised version was clearer. We certainly felt the revised structure of the introduction and section 2 was a big improvement, so we hope that the clarity is not just the result of multiple readings!

With regards to style, we apologize if the tone of our response came across as overly dismissive. Obviously we do have some different opinions with regards to style, but we did not consider the specific instances in enough detail.

In response to particular comments below, we respond on having "results" in the introduction (we've moved things round a bit to avoid being jarring, but maintaining a brief description of outcomes is our preference); redundancy of text and captions (we've reduced overlap between caption and text, and trimmed several captions substantially and removed the conversational tone that was there); we've gone through the manuscript to ensure that in-line equations and expressions read fluently to us (we altered several instances where they did not).

On using acronyms, we tweaked the text where the acronyms are defined to alert a reader more clearly about their usage. If a long label is used often enough, we feel it is acceptable to use an appropriate short hand.

On using symbols in the text, in a fairly math-intensive paper, we do believe that consistent usage of defined symbols is clearer than constantly switching between words and symbols. In the revisions here we've gone through the paper and tried to make sure that the text (including symbols and in-line expressions/equations) can be read out loud and sound fluent and meaningful.

**Major Comment:**

L433-435: It should not be on the reader to determine regional variations. Furthermore, as the authors noted in their response to other comments, they have chosen to deliberately "guide a reader" through various aspects of the study but then leave this entirely to the reader. Specifically, they are now asking a reader to (i) look at a table in the appendix that breaks down various metrics as a function of size, (ii) go to Figure 2 and use the color-coding in Figure 2a paired with Figure 2b to get a sense of the regional variations in size, and (iii) to form an interpretation of what these regional variations look like. I have spent a couple minutes trying to do this and it's borderline impossible to infer anything meaningful. If regional variations are an important aspect of this study, then there should be a short summary here of the main takeaways. Even better, is to provide a table in the appendix that explicitly breaks things down by region. If the authors do not deem this to be important, then remove this reference or state that future work could investigate regional variations or something similar.

We acknowledge the reviewer's concern about the accessibility of regional information. While regional variations of disequilibrium are not the central focus of this study, we recognize that readers will likely want more detailed information about regional variability than can be interpreted from Fig. 2a and Table A1 alone. In response to this feedback, we have added Table A2 to the appendix, which presents the same results as Table A1, but instead categorized by RGI second-order region (as in Fig. 2a) and eliminates the need for cross-referencing. We have also added a brief description of the key regional variations in Appendix A to accompany Table A2.

**Specific Comments:**

L53-56: The authors have chosen to keep results in the introduction. I don't understand the rationale for doing this, so defer to the editor on whether this is appropriate or not.

**See response to next comment:**

L56-61: The stark contrast between "traditional" writing practices (e.g., ending the introduction with a statement on what each section does) and "non-traditional" writing (e.g., mixing results with the introduction, methods, etc.) is a bit jarring. This was noted in the last round of reviews, so I defer to the editor on whether this is appropriate or not.

We now realize the reviewer is highlighting that one issue was that we put our "results" ahead of the section roadmap. We've moved it so it comes as part of discussing Section 5, where it fits much more naturally, since it briefly states the main outcome of Section 5. We hope that is an improvement, and agree that it could have been jarring in the previous location.

On the general issue of whether the main outcome of the paper can be included in the introduction, we did a little looking around. There appear to be strong opinions in either direction! We found one amusing poll on the topic, with about 20% of respondents favoring either extreme. For the senior author in this study (GHR) this is the first time he's encountered this particular reaction in a reviewer comment. If there is a house style for this journal that precludes an outcomes summary in the introduction we would certainly remove it. It is our view that an introduction can be enhanced by a brief description of where the paper is taking a reader to. Our inclusion of specific numbers is to provide readers with an idea of what we mean when we say "severe disequilibrium".

L75: are absolute brackets needed? It seems preferable not to be absolute in order to show the direction of change.

We've included a sentence explaining our choice to readers. Retreats are negative numbers, and subtractions of negative numbers can cause momentary confusion as a reader has to go through the step of subtracting a bigger negative number to get a positive number. The use of absolute brackets saves the reader this step. Moreover, it might also confuse the reader that the "committed retreat" is a positive number, since retreats are negative. We now concretely state that it is the magnitude of the difference we are interested in, and that we focus on the context of warming. The use of brackets at all makes it a grouped term, which is useful since we refer to it often in the paper.

L84: remove the comma in the citation.

Nice catch! Thank you.

L97: delete the comma after the colon.

Done, thanks.

L163 or L167: If these are Order 2 RGI regions, state this explicitly.

Yes - we've changed this to "second-order regions" to be more clear (the terminology we see used by RGI).

Figure 3: is there a reason not to use "cumulative dens." To be consistent with the PDF in (a)?

We feel that the y-axis label "cumulative frac." is more easily understood than "cumulative dist."

L299-306: here are good examples of where in-line equations look odd and reduce readability. Rather than write an equation for L300-301 that f\_eq = 0.57, one could write "The median fractional equilibration is 0.57 meaning the average glacier still has ..."

We have tweaked this particular example where the in-line equation follows an adjective, which we agree is stylistically questionable. The inline equation at the start of the paragraph here is to refresh the reader on the origin of  $f_eq$  and its interpretation. We feel that its omission would impair readability, but we add "fractional equilibration ( $f_eq$ )" in the preceding sentence to further refresh the reader. Although we differ in preference around inline equations, we appreciate the reviewer's effort here and in the prior review in working to enhance the paper's readability.

We've gone through all the other in-line expressions and equations, and considered whether they read fluently within the text. We found a few where we thought improvements could be made, for example writing out both term and symbol (i.e. "fractional equilibration (f\_eq)") when it is first used in a section to refresh the reader. We also checked through a few math-heavy articles in recent issues of The Cryosphere. We don't think we are an outlier in terms of our use of in-line equations or expressions.

L301: "still has still has"

Fixed! Thank you.

L320-336: a good example of (i) the text repeating the figure caption and (ii) a considerable amount of methods in a "results" section. I understand the authors' desire to put relevant methods near results and not use a traditional layout. In my opinion, this writing style becomes redundant and breaks up the narrative. Again, I simply mention here to point out to the editor as the authors have already made their opinion clear.

The rules we try to follow are that the figure should be understandable from the caption alone, and secondly, that the main text guides readers to the features of the graph that the authors want them to appreciate. It is frustrating as a reader to have to visually root around a graph, where what you are supposed to be seeing hasn't been stated clearly. Inevitably those two rules lead to some overlap, and erring on the side of caution might cause extra repetition. Having said that, we appreciate the reviewer's reaction, and

we've gone through how we've described each figure and tried to trim unnecessary material (from both the main text and captions).

Figure 1. We use figure 1 to walk readers through the concept of fractional equilibration, and so we want to be really clear pointing out specific features in the text. We imagine a reader carefully working with figure 1 as they go through Section 2. However, we do think the caption of figure 1 was unwieldy, and have trimmed material from it.

Figures 2 and 3 seem fine to us. They are referred to mainly parenthetically in the text, and their description in the main text is appropriate. Technical information in the caption (Fig. 2, especially) is needed and not repeated in the text

Figure 4. We identified some unnecessary repetition, and removed it from the main text. Technical details in the caption are needed and appropriate, we think.

Figure 5. We think text and caption are appropriate. It is important for us to discuss the specific numbers in the text because they contain important information that we want to highlight.

Figure 6. We were able to trim a little out of the caption. It contains a nugget of interesting information that works well in there.

Figure 7. Seems good to us. Short caption. No real overlap with the main text.

Figure 8. We were able to transfer a piece of technical information into the caption, and trim some of the rest of the caption to reduce its conversational tone. The text needs to take some space to describe the scenarios because they are integral to the rest of our analysis. We changed the topic sentence of one paragraph to break up the style a bit.

Figure 9. We changed figure 9, based on both reviewers' suggestions, to include a graphic demonstration of how uncertainty in tau affects uncertainty in f\_eq. As a result, the caption is longer, but it is not repeated in the text.

L365: appear to be missing a space after scenario,

**Thank you!**

Figure 9: move to Section 5.3 when it's first used. I assume this will be done in a typset.

We've made this adjustment for the resubmission.

L384 & Figure 9: given the explicit mention of uncertainty, can you add the uncertainty onto the figure?

This was raised by both reviewers. We have added an illustration of this to Figure 9 and hope it is clear.